# Deep-ICE: The first globally optimal algorithm for minimizing 0–1 Loss in two-Layer ReLU and Maxout networks

**Xi He**[*]
School of Computer Science
Peking University
Beijing, China
xihe@pku.edu.cn

**Yi Miao**[†]
School of Computer Science
University of Birmingham
Birmingham, B15 2TT, UK
yxm296@student.bham.ac.uk

**Max A. Little**[‡]
School of Computer Science
University of Birmingham
Birmingham, B15 2TT, UK
maxl@mit.edu

## Abstract

This paper introduces the first globally optimal algorithm for the empirical risk minimization problem of two-layer maxout and ReLU networks, i.e., minimizing the number of misclassifications. The algorithm has a worst-case time complexity of $O\left(N^{DK+1}\right)$, where $K$ denotes the number of hidden neurons and $D$ represents the number of features. It can be can be generalized to accommodate arbitrary computable loss functions without affecting its computational complexity. Our experiments demonstrate that the proposed algorithm provides provably exact solutions for small-scale datasets. To handle larger datasets, we introduce a heuristic method that reduces the data size to a manageable scale, making it feasible for our algorithm. This extension enables efficient processing of large-scale datasets and achieves significantly improved performance in both training and prediction, compared to state-of-the-art approaches (neural networks trained using gradient descent and support vector machines), when applied to the same models (two-layer networks with fixed hidden nodes and linear models).

The artifacts of the Deep-ICE algorithm can be found in https://github.com/XiHegrt/DeepICE-algorithm-artifacts.

## 1 Introduction

In recent years, neural networks have emerged as an extremely useful supervised learning technique, developed from early origins in the perceptron learning algorithm for classification problems. This model has revolutionized nearly every scientific field involving data analysis and has become one of the most widely used machine learning techniques today. Our work focuses on developing *interpretable models* for high-stakes applications, where even minor errors can lead to catastrophic consequences. For example, an incorrectly denied parole may result in innocent people suffering years of imprisonment due to racial bias (Kirchner et al., 2016), poor bail decisions can lead to the release of dangerous criminals, and machine learning–based pollution models have misclassified highly polluted air as safe to breathe (McGough, 2018). In such settings, it is crucial to deploy models that are both accurate and transparent.

---

[*]Designed the core algorithms, provided theoretical proofs, conducted the main experiments, and wrote the manuscript.

[†]Implemented the CUDA version of Deep-ICE algorithm, and co-investigated the ordered generation and memory-free techniques.

[‡]Initiated the project and provided supervision and critical feedback throughout the research and writing process.

One effective way to achieve this is to identify the best interpretable model within a given hypothesis set—a task that is uniquely suited to global optimal (exact) algorithms. Two-layer networks possess *rich expressivity*, capable of representing any continuous function (Kolmogorov, 1957), while remaining *interpretable*[1] since the output is a linear combination of hidden units. Consequently, the empirical risk minimization (ERM) problem for two-layer networks with ReLU or Maxout activation functions is not only practically useful but also theoretically significant, as it provides a foundation for understanding deep networks.

However, finding the ERM solution of a neural network remains extremely challenging. Goel et al. (2020) showed that minimizing the training error of two-layer ReLU networks under squared loss is NP-hard, even in the realizable setting (i.e., determining whether zero misclassification is achievable). This result was later extended to $L^p$ loss with $0 \leq p < \infty$ (Froese et al., 2022; Hertrich, 2022). In practice, this difficulty is further compounded when optimizing discrete loss functions, such as the 0-1 loss (count the number of misclassification), since the ultimate goal typically involves comparing classification accuracy. Even in the simplest case—linear classification using a single hyperplane—the problem of minimizing discrete losses such as the 0-1 loss is NP-hard. The best-known exact algorithm for 0-1 loss linear classification has a worst-case time complexity of $O\left(N^{D+1}\right)$, where $N$ is the number of data $D$ is the number of features (He & Little, 2023).

Nevertheless, since neural networks (NNs) have finite VC-dimension (Bartlett et al., 2019), they can, in principle, be trained exactly in polynomial time (Mohri et al., 2012). The closest related work is that of Arora et al. (2016), who proposed a one-by-one enumeration strategy to train a two-layer ReLU NN to global optimality for convex objective functions. Hertrich (2022) later extended their result to concave loss functions. However, both studies provide only pseudocode and a vague complexity analysis, without publicly available implementations or empirical validation. Moreover, they do not show how to enumerate the hyperplane partitions; instead, they assume these partitions are given.

Arora et al. (2016) further claim, somewhat ambiguously, that their algorithm has a complexity of $O\left(2^K N^{DK} poly\left(N, D, K\right)\right)$ for a two-layer ReLU network with $K$ hidden neurons with respect to $N$ data points in $\mathbb{R}^D$. The term "$poly\left(N, D, K\right)$" is not explicitly defined; it refers to the complexity of solving a *convex quadratic programming problem* with $K$ and $D$ variables and $N \times K$ constraints, and is therefore polynomial in $N$, which we denote as $O(C_1 N^{C_2})$. Therefore, Arora et al. (2016)'s algorithm involves not only extremely large exponents ($D \times K + C_2$) but also formidable constant factors ($2^K \times C_1$).

As a result of the ambiguous algorithmic description and complexity analysis, the methods proposed by Arora et al. (2016) and Hertrich (2022) appear more like a *conjecture*—suggesting the existence of a polynomial-time algorithm—rather than practically executable solutions. The prohibitive complexity in both the exponent and constant terms renders their algorithms impractical even for small-scale problems. This is further highlighted by the absence of any implementation in the *eight years* since their initial publication. Moreover, their algorithms are limited to convex loss functions, while the fundamental objective of classification is to minimize the number of misclassified instances, i.e., the 0-1 loss.

Interestingly, Bai et al. (2023) show that training a ReLU network with an $L^2$-regularized convex loss objective can be reformulated as a convex program and solved using a general-purpose solver. However, a major limitation of such solvers is their unpredictable computational complexity. Moreover, Bai et al. (2023) consider a much simpler problem than optimizing the 0–1 loss—the original objective in classification—whose discrete nature makes it substantially more difficult to optimize. Empirical results from Xi & Little (2023) further demonstrate that even for the simplest network—the linear classifier—using a general-purpose solver to optimize the 0–1 loss exhibits

---

[1]Interpretability is a domain-specific notion, so there cannot be an all-purpose definition. As Rudin (2019) noted "Usually, however, an interpretable machine learning model is constrained in model form so that it is either useful to someone, or obeys structural knowledge." We claim 2-layer ReLU/Maxout networks are interpretable because: 1. **Shallow architecture enables direct inspection**, a 2-layer neural network has a simple, transparent structure. The output is just a linear combination of these hidden unit activations. 2. **Geometric interpretation of ReLU/Maxout network is clear**, with nonlinear activations like ReLU, each hidden neuron represents a hyperplane decision boundary in the input space. The network, therefore, partitions the input space into piecewise linear regions.

highly unpredictable behavior and can incur exponential complexity, even in situations where a polynomial-time solution exists.

To address these limitations, this paper introduces the *first globally optimal algorithm for minimizing 0–1 loss in two-Layer ReLU and Maxout networks*. Our contributions can be summarized as follows:

- **First optimal algorithm for 0-1 loss**. We present the first optimal algorithm for the empirical risk minimization problem of two-layer maxout and ReLU networks under the 0–1 loss. In contrast, prior method Arora et al. (2016); Hertrich (2022) are restricted to convex loss functions, which are comparatively easier to optimize than discrete losses such as the 0–1 loss. Our algorithm extends to any computable loss function by adapting the results of He & Little (2023) without increasing worst-case complexity.

- **Two versions of the DeepICE algorithm**. Existing methods (Arora et al., 2016; Hertrich, 2022) rely on hidden assumptions. In practice, generating hyperplane predictions requires substantial computation, yet their pseudocode initializes all partitions directly without such effort. Moreover, their complexity analyses are ambiguous, hindering both understanding and reproducibility. Consequently, no implementation has emerged in the eight years since their publication. In contrast, by leveraging a general formalism, our algorithm admits a concise and unambiguous definition in a single equation (1). We further provide two variants of the DeepICE algorithm: the **sequential version** (Algorithm 2) which reuses hyperplane predictions via memoization, and the **divide-and-conquer** version (Algorithm 3), which supports parallelization without inter-processor communication.

- **Improved computational complexity**. Our algorithm achieves a complexity of $O\left(2^{K-1} \times N^{DK+1} + N^D \times D^3\right)$, substantially better than the approaches of Arora et al. (2016) and Hertrich (2022), which require $O\left(2^K \times C_1 \times N^{DK+C_2}\right)$ in both the best and worst cases. In addition, our algorithm exhibits *significantly smaller constant factors*. This efficiency enables exact solutions for datasets with formidable combinatorial complexity—for example, the problem in Figure 1, which involves *122,468,448,960* configurations, can be solved within **minutes** using our CUDA implementation.

- **Robustness**. When combined with heuristics for large-scale problems, and training accuracy is significantly higher than that of SVMs or DNNs trained with gradient descent, our algorithm demonstrates strong out-of-sample performance. This result challenges the widely held belief that optimal algorithms necessarily overfit the training data.

The remainder of this paper is organized as follows. Section 2 presents our main theoretical contributions: Section 2.1 introduces the necessary background; Section 2.2 explains how geometric insights simplify the combinatorics of the problem; Section 2.3 describes the construction of an efficient recursive nested combination generator, which is the core component of the Deep-ICE algorithm; and Section 2.4 presents the fusion law for the Deep-ICE algorithm. Section 3 reports empirical results. Finally, Section 4 summarizes our contributions and outlines directions for future research.

## 2 THEORY

### 2.1 THEORY OF LISTS

**List homomorphisms** The *cons-list* is defined as $ListR(A) = [\,] \mid A : ListR(A)$; that is, a list is either an empty list $[\,]$ or a pair consisting of a head element $a : A$ and a tail $x : ListR(A)$, concatenated using the cons operator :. For example, $1 : [2, 3] = [1, 2, 3]$. This cons-list corresponds to the singly linked list data structure in imperative languages. The key difference here is that we are referring to the model of the data structure—i.e., the datatype—rather than a specific implementation. There is a corresponding *homomorphism* over the cons-list datatype, which is a *structure-preserving map* satisfying

$$
\begin{aligned}
h([\,]) &= alg_1([\,]) \\
h(a : x) &= alg_2(a, h(x))
\end{aligned}
\tag{1}
$$

where $h : ListR\,(A) \to X$. In other words, a homomorphism over a cons-list is simply a recursion that sequentially combines each element $a$ with the accumulated result $h\,(x)$ using the algebra $alg$.

Alternatively, another list model called the *join-list* is defined as $ListJ\,(A) = [\,] \mid A \mid ListJ\,(A) \cup ListJ\,(A)$. A join-list is either empty, a singleton list, or the result of joining two sublists. The join operator $\cup$ is associative, i.e., for any $x, y : ListJ$, we have: $x \cup [a] \cup y = (x \cup [a]) \cup y = x \cup ([a] \cup y)$. The corresponding homomorphism over join-lists is a structure-preserving map defined as

$$
\begin{aligned}
h\,([\,]) &= alg_1\,([\,]) \\
h\,([a]) &= alg_2\,([a]) \\
h\,(x \cup y) &= alg_3\,(h\,(x)\,, h\,(y))
\end{aligned}
\tag{2}
$$

An example of a join-list homomorphism that computes the length of a list uses the definitions $alg_1\,([\,]) = 0$, $alg_2\,(a) = 1$, and $alg_3\,(x \cup y) = h\,(x) + h\,(y)$.

**Fusion laws**   An important principle associated with both cons-list and join-list homomorphisms is the *fusion law*, stated in the following two theorems. Its correctness can be verified either by using induction (Bird & Gibbons, 2020) or universal property (Bird & De Moor, 1996). For brevity, we omit the proofs here.

**Theorem 1.** *Fusion law for the cons-list.* Let $f$ be a function and let $h$ and $g$ be two cons-list homomorphisms defined by the algebras $alg$ and $alg'$, respectively. The fusion law states that $f \circ h = g$ if

$$
f\,(alg\,(a, h\,(x))) = alg'\,(a, h\,(x))\,.
\tag{3}
$$

Similarly, the fusion condition for the join-list is defined as following.

**Theorem 2.** *Fusion law for the join-list.* Let $f$ be a function and let $h$ and $g$ be two join-list homomorphisms defined by the algebras $alg$ and $alg'$ respectively. The fusion law states that $f \circ h = g$ if

$$
f\,(alg\,((h\,(x)\,, h\,(y)))) = alg'\,(f\,(h\,(x))\,, f\,(h\,(y)))\,.
\tag{4}
$$

In point-free style[2], this can be expressed more succinctly as $f \circ alg = alg' \circ f \times f$, where $f \times g\,(x, y) = (f\,(x)\,, g\,(y))$.

Equations (3) and (4) are referred to as the *fusion condition*, which forms the basis for proving the correctness of the derived algorithm.

## 2.2 PROBLEM SPECIFICATION

Assume we are given a data list $ds = [\boldsymbol{x}_1, \boldsymbol{x}_2 \ldots, \boldsymbol{x}_N] : [\mathbb{R}^D]$, where the points are in general position (i.e., no $d + 1$ points lie on the same $(d - 1)$-dimensional affine subspace of $\mathbb{R}^D$), and $D \geq 2$. We associate each data point $\boldsymbol{x}_n$ with a true label $t_n \in \{1, -1\}$. We extend the ReLU activation function to vectors $\boldsymbol{x} \in \mathbb{R}^D$ via an entry-wise operation $\sigma\,(\boldsymbol{x}) = (\max\,(0, x_1)\,, \max\,(0, x_2)\,, \ldots, \max\,(0, x_D))$.

Now, consider a two-layer feedforward ReLU NN with $K$ hidden units. Each hidden node is associated with an affine transformation $f_{\boldsymbol{w}_k} : \mathbb{R}^{D+1} \to \mathbb{R}$, which corresponds to a homogeneous hyperplane $h_k$ with normal vector $\boldsymbol{w}_k \in \mathbb{R}^{D+1}, \forall k \in \mathcal{K} = \{1, 2, \ldots, K\}$. These $K$ affine transformations can be represented by a single affine transformation $f\,(\boldsymbol{W}_1) : \mathbb{R}^{D+1} \to \mathbb{R}^K$, where $\boldsymbol{W}_1 \in \mathbb{R}^{K \times (D+1)}$, with rows given by the vectors $\boldsymbol{w}_k$, i.e., $\boldsymbol{W}_1^T = (\boldsymbol{w}_1, \boldsymbol{w}_2, \ldots, \boldsymbol{w}_K)$. The output of the hidden layer is then passed through the ReLU activation, followed by a linear transformation $f\,(\boldsymbol{W}_2) : \mathbb{R}^K \to \mathbb{R}$, where $\boldsymbol{W}_2 = (\alpha_1, \alpha_2, \ldots, \alpha_K)$ are the weights connecting the hidden layer to the output node. Thus, the decision function $f_{\text{ReLU}}$ implemented by the network is given by

$$
f_{\text{ReLU}}\,(\boldsymbol{W}_1, \boldsymbol{W}_2) = f\,(\boldsymbol{W}_2) \circ \sigma \circ f\,(\boldsymbol{W}_1)\,.
\tag{5}
$$

Alternatively, instead of applying the ReLU activation function $\sigma$ followed by a linear transformation $f\,(\boldsymbol{W}_2)$, the rank-$K$ maxout network with a single maxout neuron, replaces both components with a maximum operator $\max_{\mathcal{K}} : \mathbb{R}^K \to \mathbb{R}$. The resulting decision function is given by

$$
f_{\text{maxout}}\,(\boldsymbol{W}_1) = \max_{\mathcal{K}} \circ f\,(\boldsymbol{W}_1)
\tag{6}
$$

---

[2]Point-free is a style of defining functions without explicitly mentioning their arguments.

Let $\mathcal{S}$ denote the *combinatorial search space*. For the ReLU and maxout networks, we define the configurations as $s_{\text{ReLU}} = (\boldsymbol{W}_1, \boldsymbol{W}_2) \in \mathcal{S}_{\text{ReLU}}$ and $s_{\text{maxout}} = \boldsymbol{W}_1 \in \mathcal{S}_{\text{maxout}}$, respectively. The ERM problem for both network types can then be formulated as the following optimization

$$s^* = \underset{s \in \mathcal{S}}{\operatorname{argmin}} E_{\text{0-1}}(s), \tag{7}$$

where $E_{\text{0-1}}(s_{\text{ReLU}}) = \sum_{n \in \mathcal{N}} \mathbf{1}\left[\operatorname{sign}\left(f_{\text{ReLU}}\left(\boldsymbol{W}_1, \boldsymbol{W}_2, \bar{\boldsymbol{x}}_n\right)\right) \neq t_n\right]$ for ReLU network, and $E_{\text{0-1}}(s_{\text{maxout}}) = \sum_{n \in \mathcal{N}} \mathbf{1}\left[\operatorname{sign}\left(f_{\text{maxout}}\left(\boldsymbol{W}_1, \bar{\boldsymbol{x}}_n\right)\right) \neq t_n\right]$ for maxout networks. In the following discussion, we primarily focus on the maxout network, as an efficient speed-up technique is available in this setting. Unless otherwise stated, when $E_{\text{0-1}}$ is used it refers to the objective function for the maxout network by default. Although our algorithm is compatible with any computable objective function, to enable future acceleration strategies, it is beneficial to restrict the choice of objective to be a monotonic linear function of the form: $E_{\text{0-1}}(s_{\text{ReLU}}) = \sum_{n \in \mathcal{N}} L\left(\bar{\boldsymbol{x}}_n, t_n\right)$, such that $L\left(\bar{\boldsymbol{x}}_n, t_n\right) \geq 0$.

**An exhaustive search specification** Due to the *distributivity* of the ReLU activation function—that is, $\max(0, ab) = a\max(0, b)$, for $a \geq 0$—the decision function introduced by the two-layer ReLU network (5) can be rewritten as

$$f_{\text{ReLU}}\left(\boldsymbol{W}_1, \boldsymbol{W}_2, \boldsymbol{x}\right) = \sum_{k \in \mathcal{K}} \alpha_k \max\left(0, \boldsymbol{w}_k \bar{\boldsymbol{x}}\right) = \sum_{k \in \mathcal{K}} z_k \max\left(0, |\alpha_k| \, \boldsymbol{w}_k \bar{\boldsymbol{x}}\right), \tag{8}$$

where $\bar{\boldsymbol{x}} = (\boldsymbol{x}, 1) \in \mathbb{R}^{D+1}$ and $z_k \in \{1, -1\}$.

Similarly, the point-wise definition of the rank-$K$ maxout neuron are defined as

$$f_{\text{MO}}\left(\boldsymbol{W}_1, \boldsymbol{x}\right) = \max_{k \in \mathcal{K}}\left(\boldsymbol{w}_k \bar{\boldsymbol{x}}\right) \tag{9}$$

The decision function for a two-layer maxout network are simply the linear combination of maxout neurons: $f_{\text{maxout}}\left(\boldsymbol{W}_1, \boldsymbol{W}_2, \boldsymbol{x}\right) = \sum_{k \in \mathcal{K}} \alpha_k\left(f_{\text{MO}}\left(\boldsymbol{W}_1, \boldsymbol{x}\right)\right)$.

From a combinatorial perspective, the direction of the normal vector does not affect the geometric definition of its associated hyperplane. Therefore, equations (8) and (9) indicate that the decision boundary of a *two-layer ReLU* or a single *rank-$K$ maxout* neuron are fundamentally governed by a $K$-combination of hyperplanes, and then combinations of hyperplanes are composed again to form deep neural network. Although the set of all possible hyperplanes in $\mathbb{R}^D$ appears to exhibit infinite combinatorial complexity—since each hyperplane is parameterized by a continuous-valued normal vector $\boldsymbol{w}_k$—the finiteness of the dataset imposes a crucial constraint: **only a finite number of distinct data partitions** can be induced by these hyperplanes. This observation introduces a natural notion of **equivalence classes** over the space of hyperplanes, where two hyperplanes are considered in the same equivalence class if they induce the same partition over the dataset.

Indeed, according to the 0-1 loss linear classification theorem given by He & Little (2023), when optimizing the 0-1 loss (i.e., minimizing the number of misclassified data points), a hyperplanes in $\mathbb{R}^D$ an be characterized as the $D$-combinations of data points. Specifically, each critical hyperplane corresponds to the affine span of $D$ data points, leading to a total of $\begin{pmatrix} N \\ D \end{pmatrix} = O\left(N^D\right)$ possible hyperplanes. This result implies that although the parameter space is continuous, the effective combinatorial complexity of the 0-1 loss classification problem is polynomial in $N$ (for fixed $D$). Each two-layer network with $K$ hidden neurons induces up to $2^K$ distinct partitions of the input space, determined by $2^K$ possible directions of the normal vectors. These configurations can be encoded as a length-$K$ binary assignment $asgn = (a_1, \ldots a_K) \in \{1, -1\}^K$. Accordingly, a two-layer ReLU or maxout network can be characterized by the pair $cnfg = (nc, asgn) : \left(NC, \{1, -1\}^K\right)$, where $nc : NC = \left[\left[\mathbb{R}^D\right]\right]$ denotes a nested combination, representing a $K$-combination of hyperplanes.

Thus, the combinatorial search space of a two-layer NN, denoted $\mathcal{S}(N, K, D)$ consists of the *Cartesian product* of all possible $K$-combinations of hyperplanes and the $2^K$ binary assignments. A provably correct algorithm for solving the ERM problem of the two-layer network can be constructed by exhaustively exploring all configurations in $\mathcal{S}(N, K, D)$ and selecting the network that minimizes the 0-1 loss. This procedure is formally specified as

$$DeepICE(D, K) = min_{\text{0-1}}(K) \circ eval(K) \circ cp(basgns(K)) \circ nestedCombs(D, K) \tag{10}$$

where $DeepICE(D, K) : [\mathbb{R}^D] \to \left(NC, \{1, -1\}^K\right) \times Css \times NCss$, and $NCss = [[[[\mathbb{R}^D]]]]$ and $Css = [[[\mathbb{R}^D]]]$, represent nested combinations and combinations, respectively. For the parallelization concerns, $DeepICE(D, K)$ returns not only the optimal configuration for the input dataset $\mathcal{D}$ but also the intermediate representations $NCss$ and $Css$. In the specification above, the input list $xs : [\mathbb{R}^D]$ is left implicit. The function $DeepICE(D, K, ds)$ generates *all possible* $K$-combinations of hyperplanes ($K$-hidden neuron networks) by and $basgns(K)$ produces all binary sign assignments of length $K$. These are combined using the *Cartesian product* operator $cp(x, y) = [(a, b) \mid a \leftarrow x, b \leftarrow y]$. Each resulting network is then evaluated by $eval(K)$, which computes the objective value by considering all $2^K$ possible orientations of the hyperplanes and selecting the best. Finally, $min_{0\text{-}1}(K)$ selects the configuration that minimizes the 0-1 loss.

In *constructive algorithmics* community (Bird & De Moor, 1996), programs are initially defined as provably correct specifications, such as—(10)—from which efficient implementations are derived using algebraic laws like fusion. Efficiency arises both from applying fusion transformations and from designing efficient generators. To the best of our knowledge, no prior work has explored generators for nested combinations. Moreover, fusion requires that the generator be a recursive homomorphism—such as a cons-list or join-list homomorphism. This precludes the *non-recursive*, one-by-one generation approach of Arora et al. (2016) which offers opportunity for the application of acceleration techniques.

The key contribution of this paper is the development of an efficient recursive nested combination generator, $nestedCombs(D, K, xs)$, defined over a join-list homomorphism, making it amenable to fusion. The generator is tailored for efficient vectorized and parallelized implementations, making it ideal for GPU execution. We further demonstrate that $min_{0\text{-}1}$, $eval$, and $cp$ are all fusable with this generator. Additionally, the algorithm eliminates the need for an initialization step to pre-store all hyperplanes and continuously produces candidate solutions during runtime, allowing approximate solutions to be obtained before the algorithm completes.

## 2.3 An efficient nested combination generator join-list

The first step for constructing an efficient nested combinations generator requires the design of an efficient $K$-combination generator first. Previously, He & Little (2024) proposed an efficient combination generator, $kcombs$, based on a join-list homomorphism, which we extend to develop a nested combination generator.

The *nested combination-combination* generator is specified as following

$$nestedCombs(D, K) = \langle setEmpty(D), kcombs(K) \circ !!(D) \rangle \circ kcombs(D) \tag{11}$$

where $\langle f, g \rangle (a) = (f(a), g(a))$, and $!!(D, xs)$ denotes indexing into the $D$th element of the list $xs$. Equation (11) has the type $nestedCombs : Int \times Int \times [\mathbb{R}^D] \to (Css, NCss)$. It first generates all possible $D$-combinations, and then all size $D$-combinations which are then used to construct $K$-combinations. Once this process is complete, the $D$-combinations are no longer needed and are eliminated by applying $setEmpty(D)$, which sets the $D$th element of the list to an empty value.

Although the specification in (11) is correct, it requires storing the intermediate result returned by $kcombs(D, ds)$, which has a size of $O(N^D)$. Storing all these combinations is both memory-intensive and inefficient. Instead, if we can *fuse* the function $\langle setEmpty(D), kcombs(K) \circ (!!D) \rangle$ directly into the $kcombs(D)$ generator, the nested combination generator can be redefined as a single recursive process. This transformation enables incremental generation of nested combinations, eliminating the need to materialize all combinations in advance. According to the fusion law 2, this requires constructing an algebra $nestedCombsAlg$ that satisfies the following fusion condition

$$f \circ kcombsAlg(D) = nestedCombsAlg(D, K) \circ f \times f \tag{12}$$

where $f = \langle setEmpty(D), kcombs(K) \circ (!!D) \rangle$, and the definition of $kcombsAlg$ can be found in (He & Little, 2024)

The derivation of $nestedCombsAlg(D, K)$ for the empty and singleton cases is relatively straightforward. Since we assume $D \geq 2$, no nested combinations can be constructed in these cases. For the recursive case—i.e., the third pattern in the join-list homomorphism—we show that the fusion

condition holds when this third pattern of $nestedCombsAlg\,(D,K)$ is defined as

$$\Big\langle setEmpty(D) \circ KcombsAlg(K) \circ Ffst,$$
$$KcombsAlg(K) \circ \big\langle Kcombs(K)\circ!!\,(D) \circ KcombsAlg(D) \circ Ffst, \; KcombsAlg(K) \circ Fsnd \big\rangle \Big\rangle, \tag{13}$$

where $Ffst\,((a,b)\,,(c,d)) = (a,c)$, $Fsnd\,((a,b)\,,(c,d)) = (b,d)$. The proof of the fusion condition is rather complex; for readability, the complete proof is provided in Appendix A.2. Therefore, we can implement $nestedCombsAlg\,(D,K)$ as

$$nestedCombsAlg_1\,(d,k,[\,]) = ([[[\;]]]\,,[[[\;]]]])$$
$$nestedCombsAlg_2\,(d,k,[x_n]) = ([[[\;]]\,,[[x_n]]]\,,[[[\;]]]]) \tag{14}$$
$$nestedCombsAlg_3\,(d,k,(css_1,ncss_1)\,,(css_1,ncss_1)) = (setEmpty\,(D,css)\,,ncss)\,,$$

where $css = kcombsAlg\,(D,css_1,css_2)$, and $ncss$ is defined as

$$ncss = \begin{cases} [[[\;]]] & css!!\,(D) = [\,] \\ kcombsAlg\,(K,kcombsAlg\,(K,ncss_1,ncss_2)\,,kcombs\,(K,css!!\,(D))) & \text{otherwise} \end{cases}, \tag{15}$$

Thus an efficient recursive program for $nestedCombs$ is defined as the following join-list homomorphism

$$netedCombs\,(D,K,[\,]) = netedCombsAlg_1\,(D,K,[\,])$$
$$netedCombs\,(D,K,[x_n]) = netedCombsAlg_2\,(D,K,[x_n]) \tag{16}$$
$$netedCombs\,(D,K,xs \cup ys) =$$
$$\qquad netedCombsAlg_3\,(D,K,netedCombs\,(D,K,xs)\,,netedCombs\,(D,K,ys))\,,$$

Informally, the function $nestedCombsAlg\,(D,K)$ first takes as input $((Css,NCss)\,,(Css,NCss))$ which is returned by $f \times f$. The combination set is updated using the composition $setEmpty\,(D) \circ KcombsAlg\,(K) \circ Ffst$ where the first elements of the tuple are updated, and the $D$-combinations are cleared. At the same time, the function $\langle Kcombs\,(K)\circ!!\,(D) \circ KcombsAlg\,(D) \circ Ffst, KcombsAlg\,(K) \circ Fsnd \rangle$ : $((Css,NCss)\,,(Css,NCss)) \rightarrow (NCss,NCss)$ updates the combinations and nested combinations in the tuple, respectively. The newly generated $D$-combinations are then used to produce new nested combinations. Finally, the two nested combinations in the tuple are merged using $KcombsAlg\,(K) : (NCss,NCss) \rightarrow NCss$.

## 2.4 DEEP INCREMENTAL CELL ENUMERATION (DEEP-ICE) ALGORITHM AND SYMMETRY FUSION

As noted, working with the maxout network enables the application of an additional fusion principle—an extension of the symmetric fusion theorem proposed by He & Little (2023) for linear classification.

**Theorem 3.** *Symmetric fusion for maxout neuron.* Given a maxout neuron defined by $K$ hyperplane. If the predictions associated with these $K$ hyperplanes are known, then the predictions for the configuration obtained by reversing the direction of all normal vectors can be obtained directly.

*Proof.* See appendix A.1. □

The symmetric fusion theorem eliminates half of the computation, allowing us to enumerate all $2^K$ possible orientations of hyperplanes using only $2^{K-1}$ of them. Consequently, the problem 10 can be reformulated more efficiently by applying the symmetric fusion

$$DeepICE\,(D,K) = min_{\text{0-1}}\,(K) \circ eval'\,(K-1) \circ nestedCombs\,(D,K)\,,$$

where $eval'\,(K-1) = eval\,(K) \circ cp\,(basgns\,(K-1))$.

We are now ready to derive the Deep-ICE algorithm, which follows as a direct consequence of the following lemma.

**Lemma 1.** Let $DeepICEAlg$ be defined as

$$DeepICEAlg\,(D, K) = min_{\text{0-1}}\,(K) \circ eval'\,(K - 1) \circ nestedCombsAlg\,(D, K), \quad (17)$$

where $eval'\,(K - 1)$ evaluates $E_{\text{0-1}}$ for each nested combination returned by $nestedCombsAlg\,(D, K)$. Then the following fusion condition holds:

$$DeepICE\,(D, K) = f \circ nestedCombsAlg\,(D, K) = DeepICEAlg\,(D, K) \circ f \times f, \quad (18)$$

where $f = min_{\text{0-1}}\,(D) \circ eval'\,(K - 1)$, which defines Algorithm (1).

See Appendix A.3 for detailed proof. Algorithm (17) has a worst-case complexity of $O\left(N^{DK+1}\right)$, which is formally established in the following theorem.

**Theorem 4.** The DeepICE algorithm has a time complexity of $O\left(K \times N \times 2^{K-1} \times \left(\begin{pmatrix} N \\ D \\ K \end{pmatrix}\right) + N \times D^3 \times \begin{pmatrix} N \\ D \end{pmatrix}\right)$ which is strictly smaller than $O\left(N^{DK+1}\right)$, and a space complexity of $O\left(\left(\begin{pmatrix} N \\ D \\ K-1 \end{pmatrix}\right) \times K + \begin{pmatrix} N \\ D-1 \end{pmatrix} \times N\right)$, which is strictly smaller than $O\left(N^{D(K-1)}\right)$.

See Appendix A.5 for detailed proof.

In practice, we provide two implementations for (17) (see A.4). The sequential version enables two techniques that substantially improve memory efficiency and runtime performance. The D&C version, which builds upon the sequential definition, supports embarrassingly parallel execution.

Figure 2 show that the empirical wall-clock runtime of our algorithm aligns with our worst-case complexity analysis.

**Generalization to deep neural networks** Our algorithm generalizes naturally to deep neural networks. Deeper networks can be viewed as compositions of hidden neurons from preceding layers, where linear combinations of these neurons form the predictions of the subsequent layer. Hence, each layer is essentially a function of the predictions generated in the layer before it. Suppose the $i$-th hidden layer contains $K_i$ hidden nodes. Computing all possible predictions for this layer has complexity $O\left(N^{D \times K_1 \times K_2 \times K_3 \ldots \times K_i}\right)$. For instance, the optimal solution of a three-layer network is a nested-nested combination, while a four-layer network corresponds to a nested-nested-nested combination. Solving a three-layer network requires complexity $O\left(N^{D \times K_1 \times K_2}\right)$. Consequently, obtaining exact solutions for deeper networks is practically infeasible due to combinatorial explosion.

One way to mitigate this challenge is to train a deep network greedily, where the computation of the second hidden layer depends only on the first. In this case, the complexity becomes $O\left(N^{D \times K_1} + K_1^{K_2} + K_2^{K_3} \ldots + K_{i-1}^{K_i}\right)$ for network with $i$ layers. Under this scheme, regardless of depth, the overall complexity is dominated by that of the first hidden layer.

## 3 EMPIRICAL ANALYSIS

We evaluate the performance of our Deep-ICE algorithm against two baselines: support vector machines (SVMs) and an identical neural network architecture trained using Adam algorithm, referred to as MLP. The MLP is optimized with binary cross-entropy loss with logits, using the entire training dataset as a single batch in each epoch. The evaluation is conducted across 11 datasets from the UCI Machine Learning Repository. Since we assume data are in general position, which requires affine independence of the data, we remove duplicate entries and add a zero mean Gaussian noise (standard deviation $1 \times 10^{-8}$, small enough that it does not affect the results of SVM and MLP) to each dataset. All experiments were conducted on a single GeForce RTX 4060 Ti GPU.

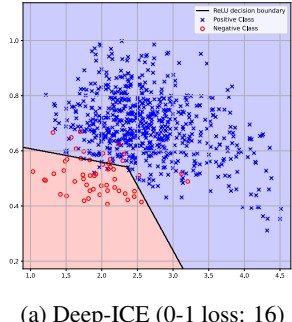

(a) Deep-ICE (0-1 loss: 16)

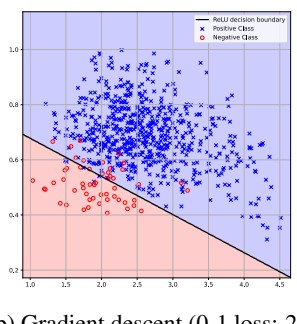

(b) Gradient descent (0-1 loss: 25)

Figure 1: The global optimal solution of a rank-2 maxout network with one neuron on a real-world dataset containing $N = 704$ data items in $\mathbb{R}^2$.

Table 1: Five-fold cross-validation results on the UCI dataset. We compare the performance of our Deep-ICE algorithm—trained either with the coreset selection method or directly by Deep-ICE algorithm (marked by *)—against approximate methods: SVM and a maxout network trained via gradient descent (denoted as MLP). Results are reported as mean 0–1 loss over training and test sets in the format: Training Error / Test Error (Standard Deviation: Train / Test). The best-performing algorithm in each row is highlighted in bold.

| Dataset | $N$ | $D$ | Deep-ICE (%) ($K = 1$) | Deep-ICE (%) ($K = 2$) | Deep-ICE (%) ($K = 3$) | SVM (%) | MLP (%) ($K = 1$) | MLP (%) ($K = 2$) | MLP (%) ($K = 3$) |
|---|---|---|---|---|---|---|---|---|---|
| Ai4i | 10000 | 6 | 97.45/97.40 (0.10/0.36) | **97.90/97.82** (0.01/0.35) | 97.71/97.71 (0.10/0.25) | 96.64/96.48 (0.11/0.44) | 97.01/96.90 (0.11/0.40) | 97.20/97.02 (0.18/0.39) | 97.56/97.55 (0.13/0.46) |
| Caesr | 72 | 5 | *74.55/82.67 (7.18/16.11) | **89.45/88.00** (4.21/9.80) | 84.36/86.67 (7.51/5.96) | 72.00/57.33 (7.14/6.80) | 71.64/62.67 (6.76/6.80) | 76.36/56.00 (6.19/9.04) | 81.82/60.00 (1.15/11.93) |
| VP | 704 | 2 | *96.94/**97.59** (0.44/1.46) | 97.76/**97.59** (0.41/1.65) | **97.80**/97.45 (0.44/2.07) | 96.77/97.02 (0.44/2.01) | 96.63/96.74 (0.50/2.13) | 96.77/97.02 (0.73/1.64) | 96.63/96.74 (0.50/2.13) |
| Spesis | 975 | 3 | *94.47/92.88 (0.10/0.61) | **96.43**/95.26 (0.49/1.82) | 96.24/**95.36** (0.22/1.62) | 94.46/92.43 (0.10/0.38) | 94.46/92.43 (0.10/0.38) | 94.46/92.55 (0.10/0.51) | 94.46/92.43 (0.10/0.38) |
| HB | 283 | 3 | *77.18/75.44 (0.45/2.48) | 80.11/77.19 (0.74/2.48) | **80.85/78.53** (1.02/3.57) | 72.40/71.23 (0.66/2.08) | 72.82/74.80 (0.41/2.01) | 75.34/75.26 (0.86/2.51) | 75.97/73.92 (0.18/2.08) |
| BT | 502 | 4 | *77.13/76.36 (1.46/2.71) | 79.59/**77.98** (0.62/3.38) | 79.36/**77.98** (0.59/2.88) | 75.09/70.14 (0.51/0.76) | 76.17/73.54 (1.05/3.64) | 76.11/73.54 (1.01/2.06) | 76.29/75.45 (1.02/2.29) |
| AV | 2342 | 7 | 89.89/88.52 (0.33/1.56) | **90.34/89.04** (0.15/1.29) | 89.77/88.76 (0.33/1.75) | 87.16/87.26 (0.31/1.24) | 86.92/87.20 (0.24/0.71) | 87.18/86.88 (0.24/0.66) | 87.63/87.31 (0.44/0.73) |
| SO | 1941 | 27 | **77.77/76.03** (0.43/0.83) | 77.13/75.33 (0.81/1.32) | 76.66/74.95 (0.74/1.38) | 73.67/70.80 (0.52/2.05) | 74.81/72.13 (0.44/1.63) | 77.09/71.71 (0.26/1.66) | 78.33/74.68 (0.40/2.31) |
| DB | 1146 | 9 | 78.78/79.69 (0.41/0.69) | 83.60/**81.37** (0.43/2.52) | **83.88**/81.32 (0.98/2.23) | 69.72/67.62 (0.65/2.86) | 76.13/74.77 (0.41/2.01) | 77.64/76.19 (0.65/1.06) | 77.85/75.11 (0.89/0.72) |
| RC | 3810 | 7 | 93.88/92.45 (0.28/1.02) | 93.91/**93.10** (0.24/1.02) | **93.94**/92.98 (0.21/0.98) | 93.05/91.75 (0.25/1.12) | 93.30/92.10 (0.28/1.07) | 93.30/92.15 (0.30/1.15) | 93.30/92.12 (0.29/1.13) |
| SS | 51433 | 3 | 86.57/**86.72** (0.03/0.15) | **86.60/86.72** (0.04/0.16) | 86.59/86.70 (0.03/0.11) | 82.77/82.75 (0.06/0.22) | 79.73/79.73 (0.15/0.20) | 79.65/79.65 (0.18/0.16) | 79.48/79.73 (0.07/0.04) |

**Exact solution vs. gradient descent** Figure 1 illustrates the ERM solution and the gradient descent outcome for a rank-2 maxout network with one maxout neuron. Previously, Xi & Little (2023) reported 0-1 losses of 19 and 23 for the global optimal linear model and the SVM, respectively, on this dataset. In contrast, ERM solution obtained by $DeepICE$, achieves only 16 misclassifications, compared to 25 for the same architecture trained via gradient descent. Notably, despite a rank-2 maxout neuron involves two hyperplanes, the gradient-based solution uses only one; the second hyperplane lies outside the data region and does not contribute to predictions.

**Exact solution over coresets** Exact solutions typically require an exhaustive exploration of the configuration space. Achieving exact optimality on training data is often unnecessary, as such solutions may not generalize well to out-of-sample data.

Instead, generating multiple high-quality candidate solutions enables selection based on validation or test performance. For example, SVMs provide tunable hyperparameters to generate alternative

models, while gradient-based MLPs yield varied solutions via different random seed initializations. However, both approaches require computationally expensive retraining to explore alternatives, often without principled guidance. Attempts to automate this process frequently rely on strong probabilistic assumptions that rarely hold in practice (Shahriari et al., 2015; Klein et al., 2017) or employ empirical heuristics (Liao et al., 2022; Wainer & Fonseca, 2021; Duan et al., 2003), resulting in substantial computational waste due to redundant retraining.

A common approach to address this issue in studies of exact algorithms is to use multiple random initializations. However, this approach often becomes ineffective as data scales increase. Each run typically uses a manually set time limit, but this still results in redundant retraining. To address these challenges, we propose a coreset selection method, detailed in Algorithm 4. Instead of computing the exact solution across the entire dataset, which is computationally infeasible for large $K$ and $D$, our approach identifies the exact solution for the most representative subsets. By shuffling the data, the input will unlikely be the ordering that is pathological i.e., one where the optimal solution is obtained only at a late stage of the recursive process in the Deep-ICE algorithm. This method can effectively explore thousands of candidate configurations in the coresets that have lower training accuracy than SVMs and MLPs. In our experiments, we trained a two-layer maxout network using the algorithmic process described in 4. In 5-fold cross-validation tests, our method demonstrated significantly better performance. These results consistently outperformed those of SVMs and the same maxout network trained with gradient descent.

Due to the ability to generate an extensive number of candidate solutions, we observed several interesting findings in our experiments. Although extensive prior research suggests that the maximal-margin (MM) classifier (i.e., SVM) offers theoretical guarantees for test accuracy (Mohri et al., 2012), we found that the MM classifier does not always perform as expected. Specifically, we did not find clear evidence that the MM classifier consistently achieves better out-of-sample performance. A more detailed analysis is provided in Appendix A.7.

Furthermore, Karpukhin et al. (2024) proposed an interesting framework that introduces stochasticity into the model's output and optimizes the expected accuracy, allowing gradient-based methods to directly optimize accuracy rather than surrogate losses. However, despite being named EXACT, the method is actually short for "EXpected ACcuracy opTimization" and is therefore a stochastic approach rather than a deterministic exact algorithm. We include a comparison with their framework in Appendix A.6, which shows that it outperforms MLPs trained with surrogate losses.

Additionally, the wall-clock runtime comparison between EXACT and MLP is provided in A.6.

## 4 DISCUSSION AND CONCLUSION

In this paper, we present the first algorithm for finding the globally minimal empirical risk of two-layer neural networks under 0–1 loss. The algorithm achieves polynomial time and space complexity for fixed $D$ and $K$. The DeepICE algorithm is specifically designed to optimize both efficiency and parallelizability. Even without bounding techniques to accelerate computation, our implementation demonstrates strong performance: it can handle over $1 \times 10^{11}$ configurations within minutes, highlighting the intrinsic efficiency of our algorithm independent of any bounding methods. Incorporating additional bounding techniques in future research could further enhance its scalability.

Another key contribution of this paper is the empirical evidence that optimal solutions do not necessarily overfit the data. Our out-of-sample tests indicate that solutions trained using our method, which achieve significantly higher training accuracy than SVMs or two-layer neural networks, still perform well on unseen data when model complexity is properly controlled. This finding points to a promising avenue for applying our algorithm to problems where both interpretability and model complexity are critical.

## REPRODUCIBILITY STATEMENT

To facilitate reproducibility, we provide **three** versions of our algorithm: a *recursive version*, a *divide-and-conquer version*, and a *sequential definition* in Appendix A.4. The recursive version is written clearly in a functional style and can be executed in a functional programming language with minimal syntactic adjustments, allowing the algorithm to run with no ambiguity. In addition,

imperative implementations in both Python and CUDA are included in supplementary materials, along with all datasets used in our experiments. Enabling independent verification and replication of the results reported in this paper.

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

## A   PROOFS

### A.1   SYMMETRIC FUSION FOR MAXOUT NETWORK

**Theorem 5.** *Symmetric fusion for maxout network.* Given a maxout network defined by $K$ hyperplane (neurons).If the predictions associated with this configuration of $K$ hyperplanes are known, then the predictions for the configuration obtained by reversing the direction of all normal vectors can be obtained directly from the original hyperplanes, without explicitly recomputing the predictions for the reversed hyperplanes.

*Proof.* Consider a maxout network defined by $K$ hyperplanes $\mathcal{H} = \{h_k \mid k \in \mathcal{K} = \{1, 2, \ldots, K\}\}$, where each hyperplane $h_k$ is defined by a normal vector $\boldsymbol{w}_k : \mathbb{R}^D$. Together these hyperplanes define a decision function $f_{\boldsymbol{W}_1, \boldsymbol{W}_2}(\boldsymbol{x})$. Equation (10) implies that a data item $\boldsymbol{x}$ is predicted to negative class by $f_{\boldsymbol{W}_1, \boldsymbol{W}_2}(\boldsymbol{x})$ if and only it lies in the negative sides of all hyperplanes in $\mathcal{H}$, because $f_{\boldsymbol{W}_1, \boldsymbol{W}_2}(\boldsymbol{x})$ will return positive as long as there exists a $k$ such that $\boldsymbol{w}_k \boldsymbol{x} \geq 0$. Therefore, the prediction labels of the two-layer NN $\boldsymbol{y}_{\text{maxout}}$ consists of the union of positive prediction labels for each hyperplane $h_k$, and the remaining data item, which lies in the negative side with respect to all $K$ hyperplanes will be assigned to negative class. class. In other words, if we denote $\boldsymbol{y}^+$ and $\boldsymbol{y}^-$ as the positive and negative prediction indexes of $\boldsymbol{y}$ respectively, then we have

$$\boldsymbol{y}^+_{\text{maxout}} = \bigcup_{k \in \mathcal{K}} \boldsymbol{y}^+_k$$
$$\boldsymbol{y}^-_{\text{maxout}} = \mathcal{D} \backslash \boldsymbol{y}^+_{\text{maxout}}$$

(19)

where $\backslash$ is defined as the set difference and $\bigcup_{k \in \mathcal{K}} \boldsymbol{y}^+_k$ denote the union of $\boldsymbol{y}^+_k$, $k \in \mathcal{K}$. For instance, if $\boldsymbol{y}_1 = (1, 1, -1, -1)$ and $\boldsymbol{y}_2 = (-1, 1, 1, -1)$, then $\boldsymbol{y}^+_1 = \{1, 2\}$ and $\boldsymbol{y}^+_2 = \{2, 3\}$, thus $\boldsymbol{y}^+_1 \cup \boldsymbol{y}^+_2 = \{1, 2, 3\}$

For a two-layer maxout NN, the data points can be classified into three categories based on their relationship to the $K$ hyperplanes defined by the $K$ hidden neurons:

1. Data points that lie in the region where all $K$ hyperplanes are on the positive side.

2. Data points that lie in the region where all $K$ hyperplanes are on the negative side.

3. Data points that lie in the region where some hyperplanes are on the positive side and others are on the negative side.

If we reverse the orientation of all $K$ hyperplanes in $\mathcal{H}$, i.e., $\boldsymbol{w}_k = -\boldsymbol{w}_k$. Only data points that fall into the class of the first two cases will be reversed, because the prediction labels of these data be reversed if the orientation for all hyperplanes is reversed, the classification of data points in the third category will remain unchanged. This is because (8) implies that, the prediction labels of the two-layer NN, $\boldsymbol{y}_{\text{maxout}}$, consist of the union of positive prediction labels for each hyperplane $h_k$.Therefore, reversing the direction of all hyperplanes will affect only data points $\boldsymbol{x}_n$ that lie in the positive class for all hyperplanes, ($n \in \boldsymbol{y}^+_k$, $\forall k \in \mathcal{K}$) or the negative class for all hyperplanes

$(n \in \boldsymbol{y}_k^-, \forall k \in \mathcal{K})$ will be change the label. For any other data points, there always exists at least one hyperplane that classifies them as negative. After reversing the direction of all hyperplanes, this same hyperplane will classify these points as positive, leaving their prediction labels unchanged. □

## A.2 Proof of nested combination generator

Given $nestedCombsAlg\,(D, K)$ defined as

$$
\Big\langle setEmpty(D) \circ KcombsAlg(K) \circ Ffst,
$$
$$
KcombsAlg(K) \circ \Big\langle Kcombs(K) \circ !!\,(D) \circ KcombsAlg(D) \circ Ffst,\; KcombsAlg(K) \circ Fsnd \Big\rangle \Big\rangle,
\tag{20}
$$

We need to verify the following fusion condition

$$
f \circ KcombsAlg\,(D) = nestedCombsAlg\,(D, K) \circ f \times f,
\tag{21}
$$

where $f = \langle setEmpty\,(D)\,, Kcombs\,(K) \circ !!\,(D) \rangle$. In other words, we need to prove that the following diagram commutes

$$
\begin{array}{ccc}
Css & \xleftarrow{\quad kcombsAlg(D) \quad} & (Css,\,Css) \\
{\scriptstyle f}\downarrow & & \downarrow{\scriptstyle f \times f} \\
(Css,\,NCss) & \xleftarrow{\quad nestedCombsAlg(D,K) \quad} & ((Css,\,NCss)\,,(Css,\,NCss))
\end{array}
$$

However, proving that the above diagram commutes is challenging. Instead, we expand the diagram by presenting all intermediate stage explicitly

$$
\begin{array}{ccc}
Css \xleftarrow{\qquad\qquad KCsA(D) \qquad\qquad} (Css,\,Css) \\
{\scriptstyle \langle SE(D),!!(D)\rangle}\downarrow \qquad\qquad\qquad\qquad \downarrow{\scriptstyle \langle SE(D),!!(D)\rangle \times \langle SE(D),!!(D)\rangle} \\
(Css,\,Cs) \xleftarrow{\;SE(D)\times\cup\;} (Css,(Cs,\,Cs)) \xleftarrow{\langle KCsA\,(D)\,\circ\,Ffst,\,\langle !!\,(D)\,\circ\,KCsA\,(D)\,\circ\,Ffst,\,\cup\,\circ\,Fsnd\rangle\rangle} ((Css,\,Cs)\,,(Css,\,Cs)) \\
{\scriptstyle id\times KCs(K)}\downarrow \qquad\qquad\qquad\qquad \downarrow{\scriptstyle (id\times KCs(K))\times(id\times KCs(K))} \\
(Css,\,NCss) \xleftarrow{\;SE(D)\times KcsA(K)\circ\cup\;} (Css,(NCss,\,NCss)) \xleftarrow{\langle KCsA\,(D)\,\circ\,Ffst,\,\langle KCs\,(K)\circ!!\,(D)\,\circ\,KCsA\,(D)\,\circ\,Ffst,\,KCsA\,(K)\,\circ\,Fsnd\rangle\rangle} ((Css,\,NCss)\,,(Css,\,NCss))
\end{array}
$$

where $\cup\,(a, b) = a \cup b$, and $SE$, $KCs$ and $KCsA$ are short for $setEmpty$, $Kcombs$ and $KcombsAlg$.

To prove the fusion condition, we first need to verify the two paths between $(Css, Css)$ and $(Css, Cs)$. In other words, we need to prove

$$
\langle SE\,(D)\,, !!\,(D) \rangle \circ KCsA\,(D) =
$$
$$
SE\,(D) \times (\cup \circ \langle KCsA\,(D) \circ Ffst, \langle !!\,(D) \circ KCsA\,(D) \circ Ffst, \cup \circ Fsnd \rangle \rangle) \circ (\langle SE\,(D)\,, !!\,(D) \rangle \times \langle SE\,(D)\,, !!\,(D) \rangle)
\tag{22}
$$

This can be proved by following equational reasoning

$$SE\left(D\right)\times\cup\circ\langle KCsA\left(D\right)\circ Ffst,\langle!!\left(D\right)\circ KCsA\left(D\right)\circ Ffst,\cup\circ Fsnd\rangle\rangle\circ\left(\langle SE\left(D\right),!!\left(D\right)\rangle\times\langle SE\left(D\right),!!\left(D\right)\rangle\right)$$

$\equiv\times$ absorption law

$$\langle SE\left(D\right)\circ KCsA\left(D\right)\circ Ffst,\cup\circ\langle!!\left(D\right)\circ KCsA\left(D\right)\circ Ffst,\cup\circ Fsnd\rangle\rangle\circ\left(\langle SE\left(D\right),!!\left(D\right)\rangle\times\langle SE\left(D\right),!!\left(D\right)\rangle\right)$$

$\equiv$ Product fusion

$$\langle SE\left(D\right)\circ KCsA\left(D\right)\circ Ffst\circ\left(\langle SE\left(D\right),!!\left(D\right)\rangle\times\langle SE\left(D\right),!!\left(D\right)\rangle\right),$$
$$\cup\circ\langle!!\left(D\right)\circ KCsA\left(D\right)\circ Ffst,\cup\circ Fsnd\rangle\circ\left(\langle SE\left(D\right),!!\left(D\right)\rangle\times\langle SE\left(D\right),!!\left(D\right)\rangle\right)\rangle$$

$\equiv$ Definition of $SE\left(D\right)$ and product fusion

$$\langle SE\left(D\right)\circ KCsA\left(D\right),\cup\circ\langle!!\left(D\right)\circ KCsA\left(D\right)\circ Fse\left(D\right),\cup\circ F!!\left(D\right)\rangle\rangle$$

$\equiv$ Definition of Combination

$$\langle SE\left(D\right),!!\left(D\right)\rangle\circ KCsA\left(D\right)$$

where $Fse\left(D,a,b\right)=\left(SE\left(D,a\right),SE\left(D,b\right)\right)$, $F!!\left(D,a,b\right)=\left(!!\left(D,a\right),!!\left(D,b\right)\right)$.

Note that, the equality between the third equation and the last equation is a assertion of fact, rather than a results can be proved (verified). This equivalence comes from the fact that size $K$-combinations can be constructed by joining all possible combinations of size $i$ and size $K-i$ combinations, where $0\leq i\leq K$.

Next, we prove the two paths between $\left(\left(Css,Cs\right),\left(Css,Cs\right)\right)$ and $\left(Css,NCss\right)$ are equivalent.

$$\langle SE\left(D\right)\circ KCsA\left(D\right)\circ Ffst,KcsA\left(K\right)\circ\cup\circ\langle KCs\left(K\right)\circ!!\left(D\right)\circ KCsA\left(D\right)\circ Ffst,KCsA\left(K\right)\circ Fsnd\rangle\rangle\circ$$
$$\left(id\times KCs\left(K\right)\right)\times\left(id\times KCs\left(K\right)\right)$$

$\equiv$ Product fusion,$f\times g=\langle f\circ Ffst,f\circ Fsnd\rangle$, $FKCssnd\left(D,\left(a,b\right),\left(c,d\right)\right)=\left(KCs\left(D,b\right),KCs\left(D,d\right)\right)$

$$\langle SE\left(D\right)\circ KCsA\left(D\right)\circ Ffst,KcsA\left(K\right)\circ\cup\circ\langle KCs\left(K\right)\circ!!\left(D\right)\circ KCsA\left(D\right)\circ Ffst,KCsA\left(K\right)\circ FKCs\left(D\right)\rangle\rangle$$

$\equiv$ Definition of $Kcombs$

$$\langle SE\left(D\right)\circ KCsA\left(D\right)\circ Ffst,KcsA\left(K\right)\circ\cup\circ\langle KCs\left(K\right)\circ!!\left(D\right)\circ KCsA\left(D\right)\circ Ffst,KCs\left(K\right)\circ\cup\circ Fsnd\rangle\rangle$$

$\equiv$ Definition of product

$$\langle SE\left(D\right)\circ KCsA\left(D\right)\circ Ffst,KcsA\left(K\right)\circ\cup\circ KCs\left(K\right)\circ\langle!!\left(D\right)\circ KCsA\left(D\right)\circ Ffst,\cup\circ Fsnd\rangle\rangle$$

$\equiv$ Definition of $KCs$

$$\langle SE\left(D\right)\circ KCsA\left(D\right)\circ Ffst,KCs\left(K\right)\circ\cup\circ\langle!!\left(D\right)\circ KCsA\left(D\right)\circ Ffst,\cup\circ Fsnd\rangle\rangle$$

### A.3 PROOF OF FUSION CONDITION

**Lemma 2.** $DeepICEAlg$ satisfies the following fusion condition

$$DeepICE\left(D,K\right)=f\circ nestedCombsAlg\left(D,K\right)=DeepICEAlg\left(D,K\right)\circ f\times f \quad (23)$$

where $f=min_{\text{0-1}}\left(D\right)\circ eval'\left(K-1\right)$, which defines the Deep ICE algorithm 17.

*Proof.* For optimization problem, proving equality is often too strict that it rarely holds in practice. Instead, whenever a "selector" is used, we can relax the fusion condition by replacing the eqaulity as a set memership relation (Bird & Gibbons, 2020).

$$f\circ nestedCombsAlg\left(D,K\right)\subseteq DeepICEAlg\left(D,K\right)\circ f\times f \quad (24)$$

In point-wise style, this is equivalent to

$$f\circ nestedCombsAlg\left(D,K,h\left(xs\right),h\left(ys\right)\right)\subseteq DeepICEAlg\left(D,K,f\left(h\left(xs\right)\right),f\left(h\left(ys\right)\right)\right) \quad (25)$$

where $h\left(as\right)=nestedCombs\left(D,K,as\right)$.

On the left side of the set membership relation, we first update the nested combinations by merging $nestedCombs\left(D,K,ys\right)$ and $nestedCombs\left(D,K,ys\right)$ using $nestedCombsAlg$ and then select the optimal $nc$ with respect to $E_{\text{0-1}}$ by using $min_{\text{0-1}}\left(D\right)\circ eval'\left(K-1\right)$.

On the right-hand side, recall that $nestedCombs\left(D, K, as\right) : \left[\mathbb{R}^D\right] \to \left(\left[\left[C\right]\right], \left[\left[NC\right]\right]\right)$ returns all possible nested combinations ($K$-combination of hyperplanes) $ncss$, all possible combination of data items $css$ ($D$th inner list is empty) and $ncss$, and $f \circ h = DeepICE\left(D, K\right)$ is the specification of the Deep-ICE algorithm. Functions $f\left(h\left(xs\right)\right)$ and $f\left(h\left(ys\right)\right)$ select the optimal nested-combination with respect to $E_{0\text{-}1}$ from all possible nested combinations with respect to $xs$ and $ys$, call them $optcnfg_1$, and $optcnfg_2$ with respectively. Then the nested combinations are merged together and selected the new optimal configuration $optcnfg'$ by using $DeepICEAlg$. By definition, $optcnfg'$ is obtained by selection the optimal configurations from the newly generated combinations and compared with $optcnfg_1$, and $optcnfg_2$, thus the solutions on the left side of the set membership relation must include in the right-hand side of the nested combination. $\qquad\square$

### A.4 ALGORITHMS

Algorithm 4 present the recursive definition of the Deep-ICE algorithm.

---

**Algorithm 1** $DeepICE_{\text{rec}}$: DeepICE recursive definition

---

**Input**: $ds$: input data list; $D$: number of features; $K$: number of hyperplanes;

**Output**: $cnfg$ : $\left(NC, \{1, -1\}^K\right)$—Optimal nested combination with respect to $ds$; $ncss$ : $NCss$—All possible nested combinations of size less than $K$; $css$ : $Css$—All possible combinations of size less than $D$.

$DeepICE\left(D, K, [\,]\right) = nestedCombsAlg_1\left([]\right)$
$DeepICE\left(D, K, [a]\right) = nestedCombsAlg_2\left([a]\right)$
$DeepICE\left(D, K, xs \cup ys\right) = min_{0\text{-}1}\left(K\right) \circ eval'\left(K - 1\right) \circ$
$\qquad nestedCombsAlg_3\left(DeepICE\left(D, K, xs\right), DeepICE\left(D, K, ys\right)\right),$
where $nestedCombsAlg$ is defined as

$nestedCombsAlg_1\left(d, k, [\,]\right) = \left(\left[\left[\left[\,\right]\right]\right], \left[\left[\left[\,\right]\right]\right]\right)$
$nestedCombsAlg_2\left(d, k, [x_n]\right) = \left(\left[\left[\left[\,\right]\right], \left[\left[x_n\right]\right]\right], \left[\left[\left[\,\right]\right]\right]\right)$
$nestedCombsAlg_3\left(d, k, \left(css_1, ncss_1\right), \left(css_1, ncss_1\right)\right) = \left(setEmpty\left(D, css\right), ncss\right).$
where $css = kcombsAlg\left(D, css_1, css_2\right)$, and $ncss$ is defined as
$$ncss = \begin{cases} \left[\left[\left[\,\right]\right]\right] & css!!\left(D\right) = [\,] \\ kcombsAlg\left(K, kcombsAlg\left(K, ncss_1, ncss_2\right), kcombs\left(K, css!!D\right)\right) & \text{otherwise.} \end{cases}$$

---

We also provide both the pesudocode for the sequential version 2 and D&C versions 3 of the Deep-ICE algorithms.

---

**Algorithm 2** $DeepICE_{\text{seq}}$: Deep-ICE sequential definition

---

**Input**: $ds$: input data list; $D$: number of features; $K$: number of hyperplanes;

**Output**: $cnfg_{\text{opt}} : \left(NC, \{1, -1\}^K\right)$—Optimal nested combination with respect to $ds$; $l_{\text{opt}}$: optimal 0-1 loss, $hyperAsgn$: All possible predictions of hyperplanes with respect to input list; $css$: all possible nested combinations of size smaller than $D$ $ncss$: all possible nested combinations of size smaller than $K$;

1. $css = \left[[[\,]]\,, []^k\right]$ // initialize combinations

2. $ncss = \left[[[\,]]\,, []^k\right]$ // initialize nested-combinations

3. $hyperAsgn = empty\left(\left(\begin{array}{c} N \\ D \end{array}\right), N\right)$ / initialize prediction of hyperplanes as a empty $\left(\begin{array}{c} N \\ D \end{array}\right) \times N$ matrix

4. $l_{\text{opt}} = N$ //initialize optimal 0-1 loss

5. **for** $n \leftarrow range\,(0, N)$ **do**: //$range\,(0, N) = [0, 1, \ldots, N - 1]$

6.    **for** $j \leftarrow reverse\,(range\,(D, n + 1))$ **do**:

7.       $updates = reverse\,(map\,(\cup ds\,[n]\,, css\,[j - 1]))$ // the $reverse$ function is used to organize configurations in revolving door ordering

8.       $css\,[j] = css\,[j] \cup updaets$ // update $css$ to generate combinations in revolving door ordering,

9.    $hyperAsgn = genModels\,(css\,[D]\,, hyperAsgn)$ // generate positive/negative predictions for each hyperplane in $css\,[D]$

10.    $css\,[D] = [\,]$ // empty $D$-combinations after generation

11.    $C_1 = \left(\begin{array}{c} n \\ D - 1 \end{array}\right), C_2 = \left(\begin{array}{c} n \\ D \end{array}\right)$

12.    $ncss' = kcombs\,(k, C_2 - C_1)$

13.    $ncss = kcombsAlg\,(K, ncss, ncss')$

14.    $cnfg', l' = eval\,(ncss\,[K]\,, hyperAsgn)$ // evaluate to the number of misclassification for each size $K$ nested combination in $ncss\,[K]$

15.    $ncss\,[K] = [\,]$ // empty size $K$ nested-combinations after evaluation

16.    **if** $l' \leq l_{\text{opt}}$:

17.       $l_{\text{opt}} = l'$

18.       $cnfg_{\text{opt}} = cnfg'$

19. **return** $cnfg_{\text{opt}}, l_{\text{opt}}, hyperAsgn, ncss, css$

---

---

**Algorithm 3** $DeepICE_{\text{D\&C}}$: Deep-ICE divide-and-conquer definition

---

**Input**: $ds$: input data list; $D$: number of features; $K$: number of hyperplanes;

**Output**: $cnfg_{\text{opt}} : \left( NC, \{1, -1\}^K \right)$—Optimal nested combination with respect to $ds$; $l_{\text{opt}}$: optimal 0-1 loss

1. $hyperAsgn = empty \left( \left( \begin{array}{c} N \\ D \end{array} \right), N \right)$ // initialize prediction of hyperplanes as a empty $\left( \begin{array}{c} N \\ D \end{array} \right) \times N$ matrix

2. $l_{\text{opt}} = N$ //initialize optimal 0-1 loss

3. $ds_i, ds_j = splitToTwo \, (ds)$// split the data set into two half

4. parallel:

5.     $res_i = DeepICE_{\text{seq}} \, (D, K, ds_i)$ // Process first data list

6.     $res_j = DeepICE_{\text{seq}} \, (D, K, ds_j)$ // Process second data list

7. sync // Wait for both tasks to complete

8. // Retrieve results: configuration, loss, hyperplane assignments, combinations

9. $cnfg_i, l_i, hyperAsgn_i, css_i, ncss_i = res_i$

10. $cnfg_j, l_j, hyperAsgn_j, css_j, ncss_j = res_j$

11. $css, ncss = nestedCombsAlg_3 \, (D, K, (css_i, ncss_i), (css_j, ncss_j))$ // Merge: Combine nested combinations from both subsets

12. $hyperAsgn = mergeAsgn \left( hyperAsgn_i, hyperAsgn_j \right)$ // Merge hyperplane assignments

13. $cnfg', l' = eval \, (ncss \, [K], hyperAsgn)$ // Evaluate merged nested combinations for size K

14. $cnfgs = \left[ (cnfg_i, l_i), \left( cnfg_j, l_j \right), \left( cnfg', l' \right) \right]$ // Collect all configurations and their losses

15. $\left( cnfg_{\text{opt}}, l_{\text{opt}} \right) = min_{\text{0-1}} \, ([cnfgs])$ // Select configuration with minimum 0-1 loss

16. **return** $cnfg_{\text{opt}}, l_{\text{opt}}$

---

Algorithm 4 shows the structure of the coreset selection method.

---

**Algorithm 4** Deep-ICE with Coreset Filtering

1. **Input**: $ds$: input data list; $M$: Block size; $R$: number of shuffle time in each filtering process; $L$: Max-heap size; $B_{\max}$: Maximum input size for the Deep-ICE algorithm; $c \in (0, 1]$: Shrinking factor for heap size

2. **Output**: Max-heap containing top $L$ configurations and associated data blocks

3. Initialize coreset $\mathcal{C} \leftarrow ds$

4. **while** $\mathcal{C} \leq B_{\max}$ **do**:

5.     Reshuffle the data, divide $\mathcal{C}$ into $\left\lceil \frac{|\mathcal{C}|}{M} \right\rceil$ blocks $\mathcal{C}_B = \left\{ C_1, C_2, \ldots, C_{\left\lceil \frac{|\mathcal{C}|}{M} \right\rceil} \right\}$

6.     Initialize a size $L$ max-heap $\mathcal{H}_L$

7.     **for** $r \leftarrow 1$ **to** $R$ **do**:

8.         $r = r + 1$

9.         **for** $C \in \mathcal{C}_B$ **do**:

10.            $cnfg \leftarrow DeepICE\,(D, K, C)$

11.            $\mathcal{H}_L.\text{push}\,(cnfg, C)$

12.        $\mathcal{C} \leftarrow unique\,(\mathcal{H}_L)$ // *Merge blocks and remove duplicates*

13.     $L \leftarrow L \times c$ // *Shrink heap size*:

14. $cnfg \leftarrow DeepICE\,(D, K, \mathcal{C})$ // *Final refinement*

15. $\mathcal{H}_L.\text{push}\,(cnfg, \mathcal{C})$

16. **return** $\mathcal{H}_L$

---

### A.5 COMPLEXITY ANALYSIS

**Theorem 6.** The DeepICE algorithm has a time complexity of $O\left( K \times N \times 2^{K-1} \times \left( \begin{pmatrix} N \\ D \\ K \end{pmatrix} \right) + N \times D^3 \times \begin{pmatrix} N \\ D \end{pmatrix} \right)$ which is strictly smaller than $O\left( N^{DK+1} \right)$, and a space complexity of $O\left( \left( \begin{pmatrix} N \\ D \\ K-1 \end{pmatrix} \right) \times K + \begin{pmatrix} N \\ D-1 \end{pmatrix} \times N \right)$, which is strictly smaller than $O\left( N^{D(K-1)} \right)$.

*Proof.* We analyze the complexity using the sequential version of the DeepICE algorithm 2. At stage $n$, the computation of lines 5–8 has complexity $O\left( n^{D-1} \right)$, since there are at most $\begin{pmatrix} n \\ D-1 \end{pmatrix}$ new $D$-combinations in each recursive step. The computation at line 9 requires $O\left( n^{D-1} \times D^3 \times N \right)$ time. Similarly, the new nested combinations at lines 12–14 has a size $O\left( \sum_{k=1}^{K} \left( \begin{pmatrix} n \\ D-1 \\ k \end{pmatrix} \right) \times \left( \begin{pmatrix} n \\ D \\ k \end{pmatrix} \right) \right)$, which requires computations of a complexity $2^{K-1} \times N \times K$ per nested combination, as each combination must evaluate $2^{K-1}$ possible hyperplane orientations.

By Vandermonde's identity, we have

$$\sum_{k=1}^{K} \left( \begin{pmatrix} n \\ D-1 \\ k \end{pmatrix} \right) \times \left( \begin{pmatrix} n \\ D \\ k \end{pmatrix} \right) = \left( \begin{pmatrix} n \\ D-1 \end{pmatrix} + \begin{pmatrix} n \\ D \end{pmatrix} \\ k \right) = \left( \begin{pmatrix} n+1 \\ D-1 \end{pmatrix} \\ k \right) \leq (n+1)^{Dk}$$

Summing over $n = 0$ to $n = N - 1$, the total time complexity becomes

$$O \left( \sum_{n=0}^{N-1} \left( D^3 \times N \times \left( \begin{array}{c} n \\ D - 1 \end{array} \right) + K \times N \times 2^{K-1} \times \left( \begin{array}{c} \left( \begin{array}{c} n+1 \\ D-1 \end{array} \right) \\ k \end{array} \right) \right) \right).$$

$$= O \left( N \times D^3 \times \left( \begin{array}{c} N \\ D \end{array} \right) + K \times N \times 2^{K-1} \times \left( \begin{array}{c} \left( \begin{array}{c} N \\ D \end{array} \right) \\ K \end{array} \right) \right)$$

$$\leq O \left( N \times D^3 \times N^D + K \times N \times 2^{K-1} \times N^{DK} \right)$$

$$= O \left( N^{DK+1} \right)$$

For memory, at lines 10 and 15, we clear the size $D$ combinations and size $K$ nested combinations, so we only need to store smaller configurations in memory. The resulting space complexity is

$$O \left( \left( \begin{array}{c} \left( \begin{array}{c} N \\ D \end{array} \right) \\ K - 1 \end{array} \right) \times K + \left( \begin{array}{c} N \\ D \end{array} \right) \times N \right) = O \left( N^{D(K-1)} \right). \tag{26}$$

$\square$

### A.5.1  ORDERED GENERATION OF COMBINATIONS

To generate $D$-combinations of data points efficiently, we employ a technique that organizes combinations in a specific order, assigning each a unique "rank." To achieve this, a critical but small function *reverse* used at line 6 of the DeepICE algorithm 2 makes it possible. This allows $D$-combinations to be organized in "revolving door ordering" and thus combinations are represented by their rank rather than storing the combinations explicitly. This approach offers two key benefits: First, storing ranks significantly reduces memory usage, from $M \times D \times 64$ bits to $M \times \log(M)$ bits ($\log(M)$ is often representable using 32 bits in coreset selection method), where $M = \sum_{k=0}^{K-1} \left( \begin{array}{c} N \\ D \\ k \end{array} \right)$. A workspace in memory is preallocated before training to store predictions associated with these hyperplanes, thereby avoiding memory allocation overhead during runtime. Second, it enables the organization of hyperplane predictions into a $\left( \begin{array}{c} N \\ D \end{array} \right) \times N$ matrix, where each row corresponds to a unique rank. As a result, the algorithm requires only $O \left( N \times D^3 \times \left( \begin{array}{c} N \\ D \end{array} \right) \right)$ time. Moreover, storing hyperplanes in a single large matrix allows exploitation of high-throughput hardware such as Nvidia GPU Tensor Cores. Without this method, predictions would need to be recomputed for each hyperplane, requiring at least $O \left( N \times D^3 \times \left( \begin{array}{c} \left( \begin{array}{c} N \\ D \end{array} \right) \\ K \end{array} \right) \right)$ time. This strategy reduces memory usage and accelerates execution without drawbacks, and it can be extended to other problems involving nested combinatorial structures.

### A.5.2  MEMORY-FREE METHOD BY USING UNRANKING FUNCTION

Building on the first technique, the second method leverages the ordered structure of $D$-combinations to eliminate the need to store $K$-combinations. An unranking function takes the rank of a combination as input and reconstructs the corresponding $K$-combination on demand. This supports the dynamic generation of combinations for a given range of rank values, thereby circumventing memory constraints that would otherwise limit the algorithm due to insufficient storage. However, it incurs an additional computational cost of $\Theta(K)$ arithmetic operations per combination due to the unranking function. Despite this, the method often improves overall efficiency by simplifying memory management, leading to more effective implementations in practice.

However, this method has a limitation: it precludes the use of bounding techniques because $K$-combinations combinations are reconstructed on demand via unranking functions rather than stored

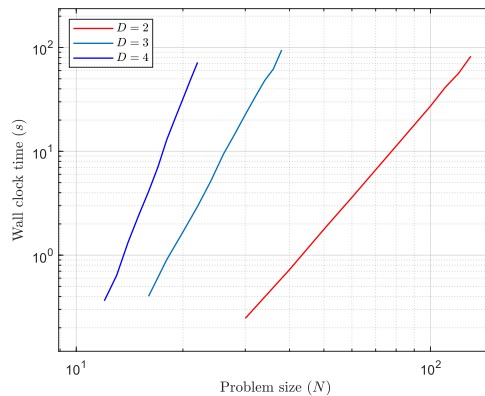 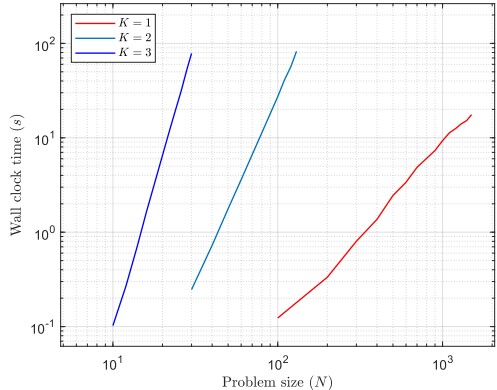

Figure 2: Empirical analysis shows that the wall-clock runtime of the DeepICE algorithm is strictly smaller than the predicted worst-case complexity $O\left(N^{DK+1}\right)$. The log-log wall-clock runtime (seconds) of DeepICE on synthetic datasets is plotted against dataset size $N$. On this log-log scale polynomial run time appears as a linear function of problem size $N$, and the slope of the line corresponds to the polynomial degree. In the left panel, the runtime curves (from left to right) correspond to $K = 2$ with $D = 2, 3, 4$, and have slopes 3.96, 6.28, and 8.88—smaller than the predicted worst-case exponents $O\left(N^4\right)$, $O\left(N^7\right)$, $O\left(N^9\right)$. In the right panel, the curves (from left to right) correspond to $D = 2$ with $K = 1, 2, 3$ respectively), and have slopes 1.91, 3.95, and 6.11—smaller than the predicted worst-case exponents $O\left(N^3\right)$, $O\left(N^5\right)$, $O\left(N^7\right)$, respectively,.

in memory. If future research requires such techniques, this approach is unsuitable, as it is challenging to identify which configurations (represented by ranks) are eliminated during algorithm execution.

### A.5.3 Empirical analysis

Figure 2 shows that the empirical running time of the DeepICE algorithm aligns with the expected worst-case complexity.

## A.6 Additional experiments

### A.6.1 Comparison with expected accuracy optimization (EXACT) framework

This Subsection we compared with the expected accuracy optimization (EXACT) method proposed by Karpukhin et al. (2024) the results is shown in table Karpukhin et al. (2024).

### A.6.2 Wall-clock run time comparison

Table 3 report the run-time comparison of between DeepICE, SVM, MLP and EXACT.

## A.7 Experiments of exhaustively exploring all solutions

For $K = 1$ case, i.e., linear case, the Deep-ICE algorithm fully explores the solution space for datasets such as Voicepath, Caesarian, Sepsis, HB, and BT. We output all solutions whose training accuracy is lower than that of the SVM. The regularization parameter for the SVM is fixed at 1 across all datasets. We deliberately avoid tuning this parameter to achieve the lowest test error, as a solution with lower test accuracy may increase training error, thereby generating more candidate solutions due to the higher training error. Adjusting the regularization parameter introduces a trade-off between training and test errors, complicating the analysis. To keep our discussion focused and consistent, we fix the regularization parameter. We summarize the empirical results in Table A.7. Using the generated solutions, we construct hyperplanes and select two representative types from each equivalence class: (1) hyperplanes passing through exactly $D$ points (direct hyperplanes), and

Table 2: Five-fold cross-validation results on the UCI dataset. We compare the performance of our Deep-ICE algorithm, with $K$ (number of hyperplanes) ranging from 1 to 3, trained either with the coreset selection method or directly (marked by *)—against Karpukhin et al. (2024)'s expected accuracy optimization (EXACT) framework. Results are reported as mean accuracy loss over training and test sets in the format: Training Error / Test Error (Standard Deviation: Train / Test). The best-performing algorithm in each row is highlighted in bold.

| Dataset | $N$ | $D$ | Deep-ICE (%) ($K=1$) | Deep-ICE (%) ($K=2$) | Deep-ICE (%) ($K=3$) | EXACT (%) ($K=1$) | EXACT (%) ($K=2$) | EXACT (%) ($K=3$) |
|---|---|---|---|---|---|---|---|---|
| Ai4i | 10000 | 6 | 97.45/97.40 (0.10/0.36) | **97.90/97.82** (0.01/0.35) | 97.71/97.71 (0.10/0.25) | 96.61/96.61 (0.01/0.02) | 96.63/96.60 (0.04/0.03) | 96.69/96.62 (0.10/0.05) |
| Caesr | 72 | 5 | *74.55/82.67 (7.18/16.11) | **89.45/88.00** (4.21/9.80) | 84.36/86.67 (7.51/5.96) | 79.50/69.24 (2.44/12.59) | 81.94/62.38 (2.65/10.03) | 87.83/64.00 (2.74/8.72) |
| VP | 704 | 2 | *96.94/**97.59** (0.44/1.46) | 97.76/**97.59** (0.41/1.65) | **97.80**/97.45 (0.43/1.71) | 92.47/92.47 (0.08/0.34) | 92.47/92.47 (0.08/0.34) | 92.47/92.47 (0.08/0.34) |
| Spesis | 975 | 3 | *94.47/92.88 (0.10/0.61) | **96.43**/95.26 (0.49/1.82) | 96.24/**95.36** (0.22/1.62) | 94.05/94.05 (0.06/0.25) | 94.05/94.05 (0.06/0.25) | 94.05/94.05 (0.06/0.25) |
| HB | 283 | 3 | *77.18/75.44 (0.45/2.48) | 80.11/77.19 (0.74/2.48) | **80.85/78.53** (1.02/3.57) | 75.70/71.39 (1.94/3.86) | 77.74/73.45 (0.55/5.89) | 78.71/71.36 (1.66/3.56) |
| BT | 502 | 4 | *77.13/76.36 (1.46/2.71) | **79.59/77.98** (0.62/3.38) | 79.36/77.98 (0.59/2.88) | 77.84/73.51 (1.31/2.80) | 78.09/73.51 (1.54/2.36) | 78.14/73.11 (1.56/3.09) |
| AV | 2342 | 7 | 89.89/88.52 (0.33/1.56) | **90.34**/89.04 (0.15/1.39) | 89.77/88.76 (0.33/1.75) | 87.18/87.18 (0.03/0.11) | 87.26/87.03 (0.19/0.37) | 87.70/87.13 (0.34/0.40) |
| SO | 1941 | 27 | **77.77/76.03** (0.43/0.83) | 77.13/75.33 (0.81/1.32) | 76.66/74.95 (0.74/1.38) | 76.33/73.11 (0.26/1.82) | 78.81/75.22 (1.68/1.77) | 79.95/75.32 (2.16/2.31) |
| DB | 1146 | 9 | 78.78/79.69 (0.41/0.69) | 83.60/**81.37** (0.43/2.52) | **83.88**/81.32 (0.98/2.23) | 73.91/70.59 (1.15/3.52) | 73.91/68.93 (3.83/4.31) | 75.94/72.42 (1.73/2.30) |
| RC | 3810 | 7 | 93.88/92.45 (0.28/1.02) | **93.91/93.10** (0.24/1.02) | 93.94/92.98 (0.21/0.98) | 93.03/92.62 (0.32/1.08) | 93.08/92.60 (0.29/1.16) | 93.08/92.76 (0.35/1.24) |
| SS | 51433 | 3 | 86.57/86.72 (0.03/0.15) | **86.60/86.72** (0.04/0.16) | 86.59/86.70 (0.03/0.11) | 86.57/86.53 (0.04/0.2) | 86.55/86.53 (0.05/0.17) | 86.56/86.53 (0.05/0.17) |

Table 3: Running time (seconds) of each algorithm, with "0.01<" denotes a time smaller than 0.01 seconds. In principle, allocating more computational resources to DeepICE yields better solutions. For comparison, we record the wall-clock time at which DeepICE first obtains a solution with lower 0–1 loss than the other methods. The reported times are the medians over three runs.

| Dataset | $N$ | $D$ | Deep-ICE (s) ($K=1$) | Deep-ICE (s) ($K=2$) | Deep-ICE (s) ($K=3$) | SVM (s) | MLP (s) ($K=1$) | MLP (s) ($K=2$) | MLP (%) ($K=3$) | EXACT (s) ($K=1$) | EXACT (s) ($K=2$) | EXACT (s) ($K=3$) |
|---|---|---|---|---|---|---|---|---|---|---|---|---|
| Ai4i | 10000 | 6 | 450.5 | 622.42 | 505.42 | 0.05 | 18.43 | 18.74 | 16.52 | 22.56 | 21.91 | 26.47 |
| Caesr | 72 | 5 | 0.26 | 0.26 | 7.10 | 0.01< | 12.27 | 12.96 | 10.11 | 23.78 | 22.92 | 29.80 |
| VP | 704 | 2 | 0.83 | 0.45 | 0.85 | 0.01< | 11.19 | 11.53 | 10.76 | 24.36 | 23.49 | 25.22 |
| Spesis | 975 | 3 | 8.00 | 0.21 | 0.41 | 0.01< | 9.56 | 10.60 | 11.68 | 25.24 | 23.01 | 29.93 |
| HB | 283 | 3 | 0.20 | 0.21 | 0.38 | 0.01< | 12.54 | 14.43 | 17.68 | 22.65 | 23.01 | 25.30 |
| BT | 502 | 4 | 0.26 | 0.36 | 0.43 | 0.01< | 13.45 | 12.34 | 15.24 | 25.46 | 24.36 | 27.99 |
| AV | 2342 | 7 | 132.51 | 294.81 | 356.51 | 0.01< | 14.53 | 14.12 | 13.21 | 22.86 | 23.02 | 28.19 |
| SO | 1941 | 27 | 762.50 | 850.42 | 543.54 | 0.04 | 12.43 | 13.23 | 14.53 | 24.12 | 25.61 | 29.36 |
| DB | 1146 | 9 | 50.43 | 20.39 | 16.77 | 0.01 | 14.78 | 16.20 | 17.43 | 26.45 | 22.57 | 27.35 |
| RC | 3810 | 7 | 423.5 | 217.26 | 611.37 | 0.02 | 15.02 | 17.02 | 14.53 | 24.68 | 22.77 | 27.21 |
| SS | 51433 | 3 | 1.13 | 3.26 | 4.21 | 9.63 | 43.19 | 73.19 | 77.43 | 25.94 | 21.86 | 26.00 |

(2) arbitrary hyperplanes computed via linear programming (LP hyperplanes). In the table, we compare the out-of-sample performance of these solutions against that of the SVM. We found no strong evidence that the maximal-margin hyperplane (SVM) consistently outperforms other hyperplanes with lower training errors. For example, in the HB dataset, an average of 8,922.2 solutions outperform the SVM in training dataset. Of these, direct hyperplanes have an average of 5,448.2 solutions, and LP hyperplanes have an average of 6,165.6 solutions, outperforming the SVM in out-of-sample test.

Table 4: Comparing the average out-of-sample accuracy in a 5-fold cross-validation. All solutions with training accuracy lower than that of the SVM are generated, and their total number is reported (Total number of solutions). Two representative hyperplanes from the equivalence classes are included: the *direct hyperplane*, which passes through exactly $D$ points, and the *LP hyperplane*, computed via linear programming. For each type, the average number of hyperplanes with out-of-sample accuracy lower than that of the SVM is also reported.

| Datasets | Total number of solutions | Direct hyperplanes | LP hyperplanes |
|---|---|---|---|
| Caesarian | 4430.2 | 2379.4 | 2913.8 |
| Voicepath | 124.2 | 55.2 | 54.8 |
| Spesis | 5.8 | 1 | 4.6 |
| HB | 8922.2 | 5448.2 | 6165.6 |
| BT | 5150.4 | 3189.8 | 3580 |

