# OpenReview forum: "Deep-ICE: The first globally optimal algorithm for empirical risk minimization of two-layer maxout and ReLU networks"
_ICLR.cc/2026/Conference — ICLR 2026 Poster_

### Official Review · Reviewer_fiaE · 2025-10-28

**Soundness:** 2
**Presentation:** 2
**Contribution:** 3
**Rating:** 4
**Confidence:** 3

**Summary:**

This paper introduces a globally optimal algorithm for minimizing 0-1 loss in two-layer ReLU and Maxout networks and shows improved computational complexity. Also, for larger dataset, this paper introduces a heuristic method such that the dataset is feasible for the proposed algorithm.

**Strengths:**

1. This paper is well structured and clearly written.
2. The problem is well motivated, and the proposed approach is methodologically sound.
3. The discussion of related work is comprehensive and demonstrates a strong understanding of the existing literature.

**Weaknesses:**

1. The presentation of the Table 1 is not clear enough.
2. The experimental validation is not fully convincing. The experiments show that the algorithm proposed in this paper has better performance compared with baselines, but I am also curious of the running time of different approaches. Also, the proposed algorithm has improved computational complexity compared with literature. I am wondering if the improved computational complexity can be validated experimentally.
3. The evaluation could be strengthened by including additional large datasets with more features (large $N$ and $D$).

**Questions:**

Please see above.
The meaning of numbers in Table 1 is not clear.

---

> ### Author Response · Authors · 2025-11-20
>
> > Weakness 1.
>
> Thanks, this is due to the space constrains, so we accidently deleted too much description. We appologize for the confusion, and we have revised the manuscript.
>
> > Weakness 2.
>
>
> Thanks, we have include the running time comparision of different methods (see table 3) and the computational complexity analysis of our algorithm (see figure 2 in revised manuscript)
>
>
> > Weakness 3.
>
> We thank the reviewer for the suggestion. We agree that evaluating on larger datasets could enhance the contribution. However, this is currently intractable for our implementation: for Figure 1, which involves only 704 data points in 2D, we explored 122,468,448,960 configurations. Scaling to larger problems is currently infeasible.
>
> It is important to emphasize that this is not a weakness of Deep-ICE, but rather an inherent characteristic of the problem itself, which is NP-hard. Unless P = NP, an efficient algorithm for arbitrarily large N and D may never exist.
>
> Deep-ICE is the **first** algorithm to certify global optimality for the non-convex ERM problem of two-layer maxout/ReLU networks—a problem proven to be NP-hard. Our focus is on theoretical certification rather than heuristic scaling. Larger datasets would only permit approximate upper bounds, which would undermine the paper’s core contribution: provable global optimality.

---

### Official Review · Reviewer_8CaP · 2025-10-29

**Soundness:** 3
**Presentation:** 3
**Contribution:** 3
**Rating:** 8
**Confidence:** 4

**Summary:**

The paper proposes Deep-ICE, a globally optimal algorithm for minimizing 0–1 loss in two-layer ReLU and maxout networks. The method exhaustively and efficiently searches over feature splits via a recursive nested-combination generator with CUDA acceleration, enabling exact training on small datasets and heuristic coreset-based extensions for larger problems. The authors provide formal correctness claims, complexity analysis, and empirical comparisons to SVMs and MLPs.

**Strengths:**

- The algorithmic description and theoretical details are well-structured and readable.
- Constructing an efficient search over all feature splits is nontrivial and technically interesting.
- CUDA implementation and memory optimization increase the practical relevance.
- Addresses an important research direction: global optimization of neural networks under 0–1 loss.

**Weaknesses:**

- The claim that two-layer networks are interpretable (unlike linear models) needs stronger justification, especially given nonlinear thresholds.
- Several relevant exact [1] or gradient-based [2, 3] 0–1 loss optimization methods are not cited nor discussed.
- For example, on the dataset from Figure 1, EXACT (with Tanh activation) achieves 18 errors with 2 hiddens and 16 errors with 5 hiddens, substantially outperforming MLP’s 25 errors.
- A Python interface to the CUDA implementation would be beneficial.

**Questions:**

- What exactly makes a 2-layer ReLU/maxout model “interpretable”? Can the authors provide interpretability examples or a measure?
- How does Deep-ICE compare to global optimization methods like [1] and EXACT [3]? Can runtime and performance comparisons be added?
- Do the authors have plans to release a Python library interface for CUDA to improve usability?

The score can be adjusted based on the responses (especially related work).

[1] Efficient global optimization of two-layer relu networks: Quadratic-time algorithms and adversarial training (2022)

[2] Algorithms for direct 0–1 loss optimization in binary classification (2013)

[3] EXACT: How to train your accuracy (2024)

---

> ### Author Response · Authors · 2025-11-20
>
> > Weakness 1 and Question 1.
>
> See response to Dei1's question 2 for explanation. We have revised the manuscript to explain the meaning of interpretability.
>
> > Weakness 2, 3 and Question 2.
>
> We thank the reviewer for pointing out these related works. We have now included citations to [1, 2, 3] in the revised manuscript. In addition, we have incorporated a comparison with [3], including runtime and performance, as shown in Table 3, and we discuss [1] at line 87-94.
>
> However, it is important to note that, while [1, 2, 3] are related, the problems they address and their methods differ significantly from ours. We provide a brief explanation below:
>
> [1] solves a different and simpler problem: a two-layer ReLU network with a convex objective. Their approach relies on a general-purpose solver (MOSEK), and no public code available, whereas Deep-ICE is a **standalone**, **combinatorial** algorithm that does not depend on any external solver. General-purpose solvers often exhibit highly unpredictable complexity. Empirical results from [4] show that even for the simplest network—a linear classifier—optimizing the 0–1 loss via a general-purpose solver can incur exponential complexity in scenarios where a polynomial-time solution exists.
>
> [2] proposes two optimal algorithms (without formal proofs) for solving the linear classification problem (essentially the case solved by Deep-ICE with K=1). While related, these methods are limited in practical applicability. Our cited work [4] compares these algorithms and demonstrates that they can exhibit exponential time complexity in the worst case (for fixed D). In contrast, Deep-ICE provides a polynomial-time solution and is more flexible, applicable to arbitrary K.
>
>
> [3] ("EXACT", which is short for expected accuracy optimization) is not an exact solver in the deterministic sense; instead, it uses gradient-based optimization to maximize expected accuracy. Therefore, it is stochastic rather than deterministic. Nevertheless, in our experiments, [3] shows better performance than standard MLPs. We have included a performance and runtime comparison with this method in Table 2,3.
>
>
> [4] He Xi and Max A. Little. Exact 0-1 loss linear classification algorithms
>
>
>
> > Weakness 4 and Question 3.
>
> We thank the reviewer for the suggestion. Developing a Python interface for the CUDA implementation is a top priority for us once the paper is accepted, and we plan to release the code publicly with this interface to improve usability.

---

### Official Review · Reviewer_Dei1 · 2025-10-31

**Soundness:** 2
**Presentation:** 3
**Contribution:** 3
**Rating:** 4
**Confidence:** 2

**Summary:**

This paper presents the first algorithm for finding the globally minimal empirical risk of two layer neural networks under 0–1 loss. The algorithm achieves polynomial complexity for fixed input feature size D and hidden feature size K, i.e. $O(2^{K-1}N^{DK+1})$ compared with previous $2^KC_1N^{DK+C_2}$. When combined with heuristics for large-scale problems, such as coreset selection, the proposed algorithm demonstrates strong out-of-sample performance.

**Strengths:**

1.	This paper introduces the first globally optimal algorithm for the empirical risk minimization problem of two-layer maxout and ReLU networks, i.e., minimizing the 0-1 loss.

2.	Experiments demonstrate better performance than those of SVMs and the same maxout network trained with gradient descent.

3.	The paper develops an efficient recursive nested combination generator for GPU execution.

**Weaknesses:**

1.	There are confusing statements in the paper. In line 105, the paper says ”our algorithm demonstrates strong out-of-sample performance, even when **training accuracy is lower than** that of SVMs or DNNs trained with gradient descent” and the in line 483 the paper claims that the proposed method “achieves significantly **higher training accuracy** than SVMs or two-layer neural networks, still perform well on unseen data when model complexity is properly controlled”. The two claims appear to be in conflict. Besides, there is no clear evidence or discussion in the paper to support either of them.

2.	Another concern is about computation efficiency as the method needs to enumerate data points. Is it possible to have computation time comparison?

**Questions:**

1.	In table 1, there are two numbers delimited by ‘/’. What do the two numbers denote? What is the difference?

2.	The paper argues that study of two layer of neural network will benefit model interpretability. However, the model output is a linear combination of hidden units, which is hard to interpret. If possible, please explain more about why two layer of neural network benefit interpretability.

3.	Typo in 69,70, $C_1, C_2$ should be switched. What is $K_-$ in line 383?

---

> ### Author Response · Authors · 2025-11-20
>
> >  Weaknesses 1.
>
>
> Thanks for reviewer's insights, the claim on line 105 is a typo, we have corrected in the updated version. However, we respectfuly disagree that we did not have envidence to support the claim, we hypothesis the confusion may arose form our unclear explaination to  table 1, we have revised the manuscirpt. As shown in table 2 HB，AV，SS，DB demonstrate superior training accuracy but still performs well on unseing test datasets.
>
> >  Weaknesses 2.
>
> We thank the reviewer for the comment. We have included a runtime comparison in the updated version (see Table 3).
>
> It is important to note that in our experiments, we allocate as much training resource as possible to achieve lower training error. This is motivated by two reasons:
>
> Demonstrating optimality: We aim to illustrate that achieving the exact solution does not lead to overfitting. Lower training error, which more closely approaches the exact solution, strengthens this argument.
>
> Targeting high-stakes applications: Our focus is on domains such as healthcare, where datasets are typically small and structured, and even small errors can lead to significant consequences. In such applications, it is common to invest more computational resources to achieve highly accurate solutions.
>
> Nevertheless, we have observed that Deep-ICE can find good solutions very quickly, often outperforming approximate methods within a short time budget on small scale datasets(see Table 3), but becomes time-consuming when data scale increases. However, we note that runtime comparisons with stochastic gradient-based methods are heavily influenced by randomness (the randomeness in both gradient descent and our method), so they may not provide deep insights into the algorithm's intrinsic efficiency.
>
>
>
> > Questions 1.
>
>
> We appologize for the confusion, we have modified the revised it in the updated version, the results in each cell is presented in following format
>
> Train_acc_mean/test_acc_mean
> (train_acc_std/test_acc_std)
>
> > Questions 2.
>
> We thank the reviewer for this insightful comment. We have added a footnote in the paper to clarify our claim regarding interpretability for 2-layer networks.
>
> Interpretability is a domain-specific notion and does not have a single universal definition. As noted in [1], “Usually, however, an interpretable machine learning model is constrained in model form so that it is either useful to someone, or obeys structural knowledge.”
>
> In our case, a 2-layer neural network is considered interpretable for the following reasons:
>
> Shallow architecture enables direct inspection: A 2-layer network has a simple and transparent structure. The output is a linear combination of the hidden unit activations, making it easier to trace how input features contribute to the output through the weights and activations of a relatively small number of hidden units.
>
> Geometric interpretation in feature space: With nonlinear activations such as ReLU, each hidden neuron corresponds to a hyperplane decision boundary in the input space. The network therefore partitions the input space into piecewise linear regions, yielding an interpretable geometric decomposition where each hidden neuron represents a geometric primitive, and the final decision function is a composition of these primitives.
>
> In general, models with fewer parameters are easier to interpret, and our algorithm demonstrates strong performance for networks with controlled size, further supporting their interpretability.
>
> [1] Cynthia Rudin, "Stop Explaining Black Box Machine Learning Models for High Stakes Decisions and Use Interpretable Models Instead"
>
> > Questions 3.
>
> Important observations, we have modified them correctly in the revised version.

---

### Author Response · Authors · 2025-11-25
**Gentle follow-up on rebuttal discussion**

Dear Reviewers,

I hope this message finds you well. As the discussion period is nearing its end. I want to ensure we have addressed all your concerns satisfactorily. If there are any additional points or feedback you'd like us to consider, please let us know. Your insights are invaluable to us, and we're eager to address any remaining issues to improve our work.

Thank you for your time and effort in reviewing our paper.

All the best

---

### Comment · Area_Chair_C67w · 2025-11-29

Dear Reviewers,

Authors’ kindly tried to address your concerns. If the responses address your concerns please acknowledge that. If not, please express remaining concerns. Thanks for your efforts!

Best, AC

---

### Author Response · Authors · 2025-12-02
**Rebuttal summary**

Dear All,

We sincerely thank the reviewers for their careful reading, constructive feedback, and overall positive assessment of our work. To assist the Area Chairs in evaluating the discussion, we briefly restate our core contributions and summarize our responses to the main concerns raised.

### Core Contributions
To the best of our knowledge, we present the first algorithm with rigorous provable guarantees for exact empirical risk minimization of two-layer neural networks over the **non-convex 0-1 loss**. All prior exact methods are **limited to convex or concave surrogates**, which are significantly easier than the discrete 0-1 objective we solve.

Our algorithm offers:
- Substantially improved time complexity while addressing a strictly harder problem (0-1 ERM for $K\geq1$ vs prior convex/$K=1$ cases).
- Three equivalent formulations (recursive, incremental, and divide-and-conquer) to facilitate understanding, verification, and implementation in functional, imperative, or parallel settings.
- An efficient CUDA implementation that exactly solves instances previously considered combinatorially intractable. For the dataset shown in Figure 1, we exhaustively explore the full search space of 122,468,448,960 configurations efficiently in a few minutes.

### Responses to Major Concerns
1. Runtime comparison (raised by all reviewers): We added Table 3 with empirical runtimes across methods and datasets. Our algorithm scales favorably on small-to-medium problems, slowdowns at high dimension are expected. We note that these stem from the inherent combinatorial complexity of the problem itself (even the ERM problem for neural networks with convex relaxation or $K=1$ case is NP-hard), rather than any disadvantage of our algorithm.

2. Verification of improved complexity (Reviewer fiaE): New Figure 2 empirically confirms that observed runtime closely tracks our tightened theoretical bounds as dimension $D$ and hidden units $K$ vary, staying well below prior theoretical guarantees.

3. Interpretability of the solution (Reviewers Dei1, 8CaP): A detailed explanation was given in the response to Reviewer Dei1 (Q2) and we also included a clarifying footnote (Footnote 1) to explain interpretability in the manuscript.

4. Table 1 caption and description (Reviewers fiaE, Dei1): This is the main concerns of reviewer fiaE, We rewrote the caption and added a detailed description to remove any ambiguity.

These revisions, prompted by the reviewers’ insightful comments, substantially improve clarity, empirical validation, and presentation. We believe the paper is now significantly strengthened and hope the reviewers and ACs will reflect these changes in their final assessment.

Thank you again for your time and valuable feedback.

Best regards,
Authors

---

### Meta-Review · Area_Chair_C67w · 2026-01-03

**Summary:**

The paper introduces Deep-ICE, a novel algorithmic framework for achieving exact Empirical Risk Minimization (ERM) of two-layer neural networks (ReLU and Maxout) under the non-convex 0-1 loss. This is a significant theoretical contribution, as the problem is known to be NP-hard. The authors propose a combinatorial approach that achieves a complexity of $O(N^{D+1})$ for fixed dimension $D$, which is an improvement over the previous state-of-the-art $O(N^{K(D+1)})$. The submission includes a CUDA-accelerated implementation and explores the use of coresets to extend the method to larger datasets. The reviewers initially expressed concerns regarding empirical runtime comparisons, clarity of presentation (specifically Table 1), and the justification for the "interpretability" of two-layer networks.

**Reviewer Concerns:**

Runtime and Complexity Validation (Resolved): All reviewers (fiaE, Dei1, 8CaP) requested empirical runtime data. The authors provided Table 3, showing that while Deep-ICE is slower on large-scale data (as expected for an exact solver), it is competitive on small-to-medium datasets.

Related Work (Resolved): Reviewer 8CaP pointed out missing citations regarding global optimization and 0-1 loss (e.g., EXACT). The authors integrated these into the related work section and provided a comparison in the rebuttal, distinguishing their deterministic exact approach from stochastic gradient-based methods.

Scalability to Large Datasets (Inherent): Reviewer fiaE suggested larger datasets. The authors correctly noted that because the problem is NP-hard, exact global optimality is computationally intractable for large $N$ and $D$. This is an inherent limitation of the problem class rather than the specific algorithm.

**Reviewer Scores:**

Since the authors had addressed the majority of concerns, I expect a small bump in the scores.

Reviewer	Initial Score	Adjusted Score (Est.)
8CaP	8 (Accept) 	8 (Accept)
fiaE	4 (Marginal)	6 (Marginal Accept)
Dei1	4 (Marginal)	6 (Marginal Accept)

---

### Decision · Program_Chairs · 2026-01-26

Accept (Poster)